# Single-atom catalysts-based catalytic ROS clearance for efficient psoriasis treatment and relapse prevention via restoring ESR1

Xiangyu Lu [1,2,8], Le Kuai[3,4,8], Fang Huang[1,5,8], Jingsi Jiang[6], Jiankun Song[6], Yiqiong Liu[6], Si Chen[1,2], Lijie Mao[1,2], Wei Peng[7], Ying Luo[3,4], Yongyong Li[6], Haiqing Dong[5] ✉, Bin Li[4,6] ✉ & Jianlin Shi [1,2] ✉

Psoriasis is a common inflammatory disease of especially high recurrence rate (90%) which is suffered by approximately 3% of the world population. The overexpression of reactive oxygen species (ROS) plays a critical role in psoriasis progress. Here we show that biomimetic iron single-atom catalysts ($FeN_4O_2$-SACs) with broad-spectrum ROS scavenging capability can be used for psoriasis treatment and relapse prevention via related gene restoration. $FeN_4O_2$-SACs demonstrate attractive multiple enzyme-mimicking activities based on atomically dispersed Fe active structures, which are analogous to those of natural antioxidant enzymes, iron superoxide dismutase, human erythrocyte catalase, and ascorbate peroxidase. Further, in vitro and in vivo experiments show that $FeN_4O_2$-SACs can effectively ameliorate psoriasis-like symptoms and prevent the relapse with augmented efficacy compared with the clinical drug calcipotriol. Mechanistically, estrogen receptor 1 (ESR1) is identified as the core protein upregulated in psoriasis treatment through RNA sequencing and bioinformatic analysis. Together, this study provides a proof of concept of psoriasis catalytic therapy (PCT) and multienzyme-inspired bionics (MIB).

Psoriasis is an inflammatory skin disorder featuring chronic, relapsing erythematous plaques covered with silvery-white scurfy scales[1–3]. The prevalence rate of psoriasis ranges from 0.47% to 6.6% in China, America, Europe, and Australia, and continues to rise[4]. Sustained inflammation and non-regulated keratinocyte proliferation, especially the burst releases of inflammatory factors in the psoriasis development, are the primary culprits involved in the pathogenesis[5,6]. Although current therapies permit the control of psoriatic inflammation, the prolonged immunosuppression, however, increases body's susceptibility to infection and skin cancer[7–9]. Additionally, psoriasis is often accompanied by other metabolic diseases, such as cardiovascular disease and diabetes, which will cause serious harms to human health. More seriously, the biggest clinical challenge for psoriasis treatment is symptom recurrence after drug discontinuation (approximately 90%)[10].

[1]Shanghai Tenth People's Hospital, Shanghai Frontiers Science Center of Nanocatalytic Medicine, Clinical Center For Brain And Spinal Cord Research, School of Medicine, Tongji University, Shanghai 200092, China. [2]Shanghai Institute of Ceramics, Chinese Academy of Sciences, Research Unit of Nanocatalytic Medicine in Specific Therapy for Serious Disease, Chinese Academy of Medical Sciences, Shanghai 200050, China. [3]Department of Dermatology, Yueyang Hospital of Integrated Traditional Chinese and Western Medicine, Shanghai University of Traditional Chinese Medicine, Shanghai 200437, China. [4]Institute of Dermatology, Shanghai Academy of Traditional Chinese Medicine, Shanghai 201203, China. [5]Key Laboratory of Spine and Spinal Cord Injury Repair and Regeneration, Ministry of Education, Tongji Hospital, School of Medicine, Tongji University, Shanghai 200065, China. [6]Shanghai Skin Disease Hospital, School of Medicine, Tongji University, Shanghai 200443, China. [7]Institute of Waste Treatment and Reclamation, College of Environment Science and Engineering, Tongji University, Shanghai 200092, China. [8]These authors contributed equally: Xiangyu Lu, Le Kuai, Fang Huang. ✉e-mail: inano_donghq@tongji.edu.cn; lib@shskin.com; jlshi@mail.sic.ac.cn

Thus, alternative therapeutic modalities for psoriasis inflammation alleviation and relapse intervention are urgently needed[11–14].

Reactive oxygen species (ROS), such as superoxide anions ($O_2^{•-}$) and hydrogen peroxide ($H_2O_2$), are modulators of inflammation that can stimulate inflammatory cytokine release, thereby stimulating the proliferation of keratinocytes[15–17]. Excessive ROS accumulation caused by oxidative/anti-oxidative imbalance and resultantly elevated oxidative damage are observed in psoriatic patients[12]. From this perspective, downregulating the ROS level at the lesion site would be highly desirable in the treatment of psoriasis. Nevertheless, ROS cannot be reduced sustainably by consuming antioxidants such as bilirubin[18,19] and monomethyl fumarate[20]. Further investigation has revealed that in response to ROS, the body has a natural anti-oxidation system, mainly composed of antioxidant enzymes, such as catalase (CAT) and superoxide dismutase (SOD), which scavenge $O_2^{•-}$ and $H_2O_2$ through catalytic reactions to maintain homeostasis[21–23]. Unfortunately, it has been discovered that these antioxidant enzymes are underexpressed in patients suffering psoriasis, which could prevent efficient ROS scavenging and even exacerbate the symptom[24]. Thus, one potential psoriasis treatment strategy is to restore redox homeostasis at the lesion by utilizing the catalytic reactions of antioxidant enzymes, thereby ameliorating inflammatory infiltration and hyperproliferation of HaCaT cells, i.e., to achieve psoriasis catalytic therapy (PCT). Nevertheless, the natural antioxidant enzymes are particularly problematic in clinical applications (e.g., single catalytic activity, low permeability, and high cost). One effective strategy is to design biomimetic catalytic materials with multiple catalytic activities by simulating the pivotal structural features of natural enzymes, which is herein defined as multienzyme-inspired bionics (MIB)[25–30].

Fortunately, single-atom catalysts (SACs) with tunable active metal centers and coordination environments provide an effective solution in mimicking the highly evolved catalytic center of natural enzymes at the atomic level[31–33]. However, most previous work has focused on mimicking the active center of a single natural enzyme. To better construct antioxidant enzyme-like catalysts, this work explores a strategy to simultaneously simulate the catalytic centers of multiple natural antioxidant enzymes.

In the antioxidant enzymes human erythrocyte CAT (PDB: 1dgf)[34] and ascorbate peroxidase (APX; PDB: 1v0h)[35], the catalytic centers are the Fe-porphyrin (Heme) structures, in which the active Fe is linked with four nitrogen atoms. Interestingly, Fe-SOD (PDB: 1avm)[36] features $FeN_3$ as an active site (Fig. 1). Consequently, we hypothesize that Fe-based SACs including Fe active site coordinated by nitrogen atoms could potentially act as multi-active natural antioxidant enzyme mimics.

In this work, we formulate a biomimetic $Fe-N_4O_2$ structure-based, atomically dispersed Fe-N-C catalyst ($FeN_4O_2$-SACs) to efficiently scavenge different types of ROS by its multiple CAT, APX, and SOD-mimicking activities. Due to their effectiveness in downregulating inflammatory factor levels, preventing keratinocyte proliferation and ROS generation, $FeN_4O_2$-SACs present significant efficacy in treating psoriasiform dermatitis, and notably, reduce the relapse of psoriasis after drug discontinuation. Furthermore, the underlying mechanism of the catalytic therapy for psoriasis is the up-regulation of estrogen receptor 1 (ESR1) in the $FeN_4O_2$-SACs treatment, which is largely reduced in the clinical samples of psoriasis and relapse patients. Taken together, this biomimetic catalyst delivers a "safe, effective, and long-term" therapeutic modality for psoriasis treatment as a typical example of multienzyme-inspired bionics (Fig. 1).

## Results
### Synthesis of $FeN_4O_2$-SACs
A schematic illustration of the synthesis of $FeN_4O_2$-SACs is provided in Fig. 2a. Briefly, Fe-doped ZIF-8 (Fe-MOF) was pyrolysed at 800°C for 3 h and then, the pyrolysate was leached with sulfuric acid to obtain $FeN_4O_2$-SACs. Analogously, N-doped porous carbon (NC) without Fe doping was prepared as a control.

Transmission electron microscopy (TEM) images show that both Fe-MOF and $FeN_4O_2$-SACs are relatively uniform in size with cubic morphologies (Supplementary Fig. 1 and Fig. 2b). Scanning electron microscopy (SEM) further displays the dodecahedral structure of $FeN_4O_2$-SACs (Supplementary Fig. 2). Notably, the sizes of $FeN_4O_2$-SACs (≈150 nm) particles are relatively smaller than those of Fe-MOF, and a pore structure appears on the surface of $FeN_4O_2$-SACs, which originated from the evaporation and overflow of Zn species during pyrolysis[37]. Surface area analysis demonstrates that $FeN_4O_2$-SACs enrich with micro- and mesopores with large specific surface area (Supplementary Fig. 3), which facilitate the exposure of catalytically active sites. The individually scattered bright dots in the aberration-corrected high-angle annular dark-field scanning TEM (AC HAADF-STEM) image confirm the atomic dispersion of Fe atoms (Fig. 2c). In parallel, the electron energy loss spectrum (EELS) and X-ray photoelectron spectroscopy (XPS) identify the presence of Fe signals (Fig. 2e and Supplementary Fig. 4), demonstrating successful Fe doping. The concentration of Fe is 0.43% as quantified by inductively coupled plasma optical emission spectrometry (ICP-OES). The HAADF-STEM image and energy-dispersive spectrum (EDS) further indicate uniform distributions of C, N, and Fe elements (Fig. 2d). To determine whether Fe-derived crystalline structures were present or not, high-resolution TEM (HRTEM) and selected area electron diffraction (SAED) were performed. As depicted in Supplementary Figs. 5 and 6, no metal crystals formed in $FeN_4O_2$-SACs, which is consistent with X-ray diffractometer (XRD) pattern (Supplementary Fig. 7 and Supplementary Discussion). Overall, we have successfully synthesized $FeN_4O_2$-SACs with isolated Fe atoms dispersed in the carbon matrix, homogeneous size, and porous structure.

### Structure of $FeN_4O_2$-SACs
The structure of $FeN_4O_2$-SACs was further investigated by various techniques. Fourier transform infrared (FTIR) spectra and Raman spectra show that $FeN_4O_2$-SACs possess a carbon structure similar to that of NC, which might be related to the low level of Fe doping (Supplementary Figs. 8 and 9 and Supplementary Discussion). Next, Mössbauer spectroscopy was used to determine the electronic state, coordination, and electron spin configuration of $FeN_4O_2$-SACs[38]. Figure 3a shows that $FeN_4O_2$-SACs display the well-documented D1 and D3 doublets attributable to Fe-N species, based on the isomer shift ($\delta_{iso}$) and quadrupole splitting ($\Delta E_Q$) values (Supplementary Table 1). The parameters of the first doublet D1 and third doublet D3 were assigned to the $Fe^{2+}N_4$ centers and low spin $Fe^{3+}N_4$ centers (heme-like), respectively[39,40]. The relative absorption area (74.4%) of D3 illustrates the significant presence of heme moieties in $FeN_4O_2$-SACs. In addition, Fe in $FeN_4O_2$-SACs is biased towards the 3+ valence state, which favors antioxidant reactions rather than peroxidase-like catalytic reactions[41,42]. Notably, no significant Fe clusters are evident in the Mössbauer spectrum.

XAFS, a powerful technique for atomic level structure characterization, was carried out to identify the chemical state and coordination environment of the Fe species[43]. Normalized X-ray absorption near-edge structure (XANES) curves verify that the pre-edge position of $FeN_4O_2$-SACs matches that of $Fe_3O_4$, indicating that the Fe valency in $FeN_4O_2$-SACs falls in between +2 and +3, similar to the results of the Mössbauer spectrum (Fig. 3b). The corresponding Fourier-transformed extended X-ray absorption fine structure (FT-EXAFS) spectra (Fig. 3c) reveal a dominant peak at 1.47 Å, which could be designated as the Fe-N(O) scattering path in the first shell. Meanwhile, a weak peak at 2.31 Å is observed in $FeN_4O_2$-SACs, which can be ascribed as Fe-N-C in the second shell[44,45]. Similar results are evidenced by the wavelet transform (WT) plot (Fig. 3d–g). By comparison with reference samples of Fe Foil, $Fe_2O_3$ and $Fe_3O_4$, $FeN_4O_2$-SACs do not display the intensity maximum corresponding to Fe-Fe, proving that all

the Fe species are present in the single-atom state. Then, least-square EXFAS fitting was performed to obtain quantitative structural parameters. As shown in Supplementary Fig. 10 and Supplementary Table 2, the Fe atoms in $FeN_4O_2$-SACs exhibit a six-fold coordination by N/O atoms and a mean Fe-N/O bond length of 1.99 Å. Since adjacent coordination elements in the periodic table (such as N and O) could not fit well in EXAFS, the presence of Fe-O coordination shells could not be excluded[44]. It was speculated that this might be the structure of Fe-$N_4O_2$ in combination with Mössbauer spectrum results and the literature[46]; and the oxygen element could derive from the atmospheric oxygen in the experiment, though slight amount of Fe-$N_5$ or Fe-$N_6$ could be present[44,46]. XPS further shows that the N types

coordinated with the Fe atoms are pyridine N, pyrrole N, and graphene N, corresponding to 398.6, 399.8, and 401.2 eV, respectively (Supplementary Fig. 11)[47]. Altogether, $FeN_4O_2$-SACs share a similar Fe-$N_4$ structure to heme, which has encouraged us to further investigate its enzymatic activity.

## SOD, CAT, and APX-like catalytic activities of $FeN_4O_2$-SACs

$O_2^{\bullet -}$ is a kind of ROS with strong oxidizing and destructive potentials. As a pivotal $O_2^{\bullet -}$ scavenger, the enzyme SOD catalyses the disproportionation of $O_2^{\bullet -}$ into $H_2O_2$ and $O_2$ (Fig. 4a)[48]. Here, the clearance of $O_2^{\bullet -}$ was verified via electron spin resonance (ESR) measurements using 5-tert-butoxycarbonyl 5-methyl-1-pyrroline N-oxide (BMPO) as

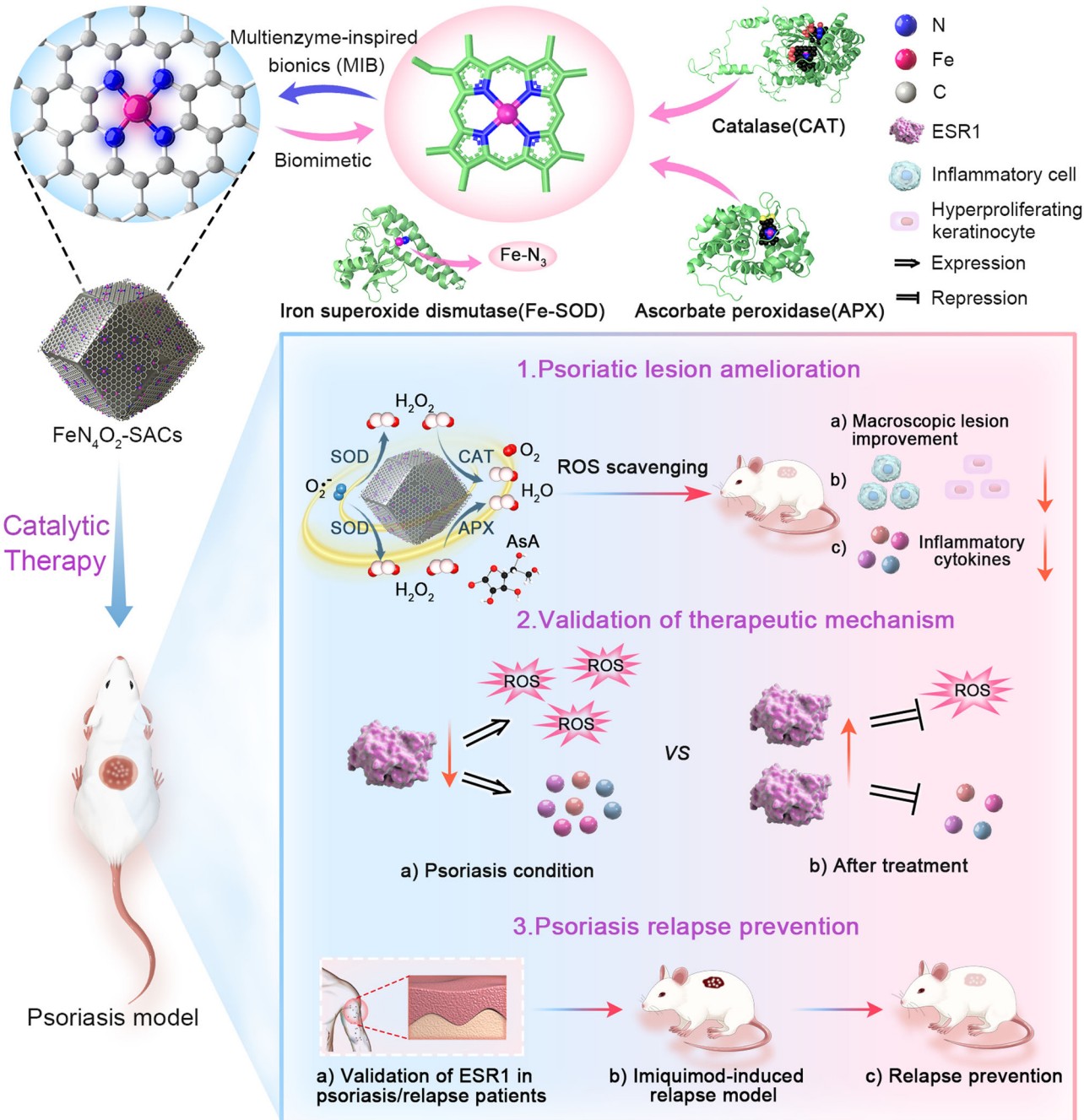

**Fig. 1 | Schematic illustration of $FeN_4O_2$-SACs for psoriasis catalytic therapy.** Inspired by the common structure feature for the activity centers of three antioxidant enzymes, $FeN_4O_2$-SACs with SOD-, CAT-, and APX-like activities are synthesized for ROS scavenging in psoriasis treatment. As expected, $FeN_4O_2$-SACs effectively ameliorate psoriatic skin lesions in vitro and in vivo. The core of the action of $FeN_4O_2$-SACs is further validated to be the upregulation of ESR1, which is downregulated in psoriasis and relapse patients. Finally, the effective prevention of relapse is achieved by $FeN_4O_2$-SACs.

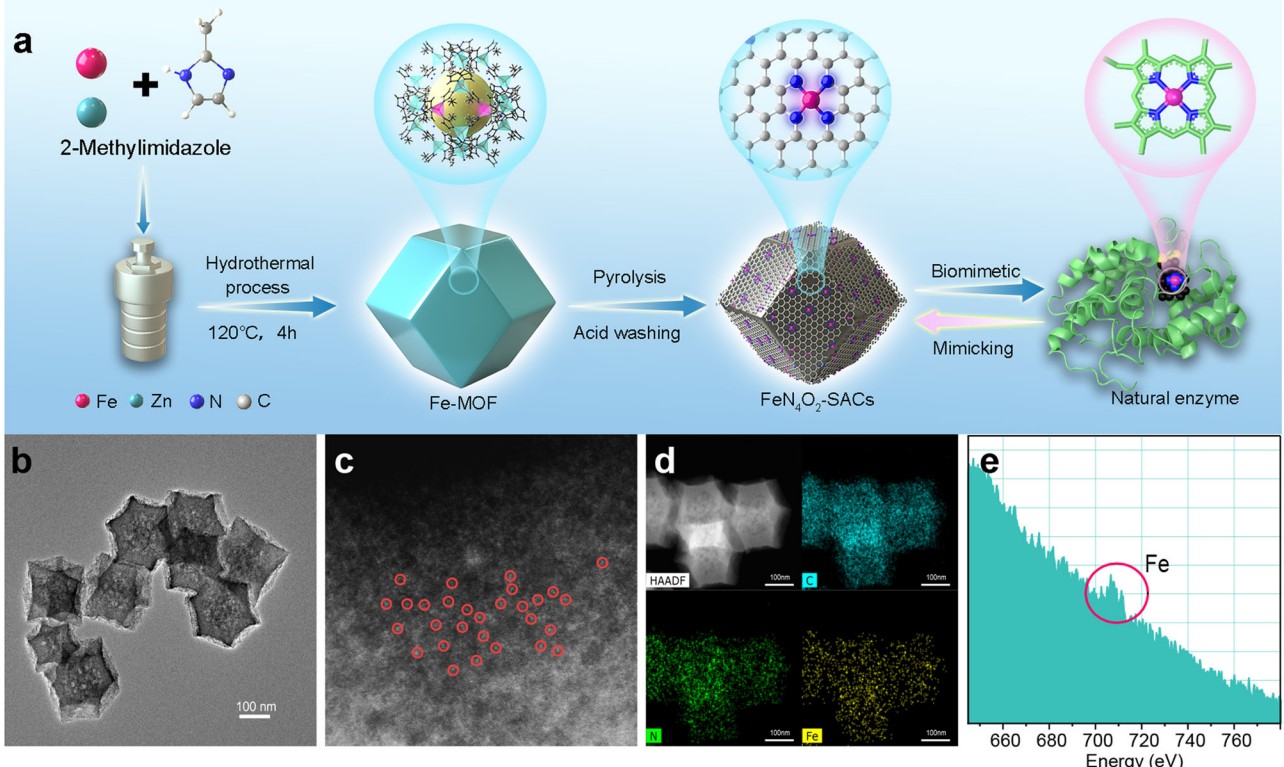

**Fig. 2 | Synthesis and characterization of FeN$_4$O$_2$-SACs. a** Synthetic process of biomimetic FeN$_4$O$_2$-SACs. **b** TEM image of FeN$_4$O$_2$-SACs. $n = 3$ samples with similar results. **c** AC HAADF-STEM image of FeN$_4$O$_2$-SACs, with single iron atoms marked using red circles. $n = 3$ samples with similar results. **d** HAADF-STEM image and corresponding EDX mappings of FeN$_4$O$_2$-SACs: C (blue), N (green), Fe (yellow). **e** EELS atomic spectrum of Fe elements in FeN$_4$O$_2$-SACs. The element-specific absorption edge is highlighted by a red circle. Source data are provided as a Source Data file.

the O$_2^{\bullet-}$ trapping reagent. ESR spectra show that O$_2^{\bullet-}$ is consumed efficiently by FeN$_4$O$_2$-SACs at pH 7.4 with the characteristic peak intensity similar to that in the presence of natural SOD, indicating that FeN$_4$O$_2$-SACs have the SOD-like activity (Fig. 4b)[49]. Likewise, this result has been demonstrated at pH 6 and 8 (Supplementary Fig. 12). Next, the SOD-like performance was quantified with a representative WST-8 colorimetric analysis. As shown in Supplementary Figs. 13 and 14, the O$_2^{\bullet-}$ scavenging rates of FeN$_4$O$_2$-SACs and natural SOD increase with increasing concentrations, and the inhibition rate of 0.13 μg/ml Fe in FeN$_4$O$_2$-SACs is almost equivalent to 0.49 μg/ml natural SOD. Finally, to evaluate the O$_2^{\bullet-}$ scavenging efficiency of FeN$_4$O$_2$-SACs, the commercial anti-inflammatory nanozymes (Mn$_3$O$_4$ and CeO$_2$) and metal single-atom catalysts (Co-SACs, Cu-SACs, Zn-SACs) were employed as controls[50–53]. The results show that FeN$_4$O$_2$-SACs exhibit the highest SOD-like activity, and the activities of Fe in FeN$_4$O$_2$-SACs are 622, 342, and 377 times those of Ce in CeO$_2$, Mn in Mn$_3$O$_4$, and Zn in Zn-SACs, respectively (Fig. 4c, Supplementary Table 3).

Although SOD can protect the organism from O$_2^{\bullet-}$ damage, H$_2$O$_2$ generated via SOD can destroy DNA, proteins and lipids[54,55]. Clearance of H$_2$O$_2$ from the body relies heavily on the enzyme CAT, which catalyzes the conversion of H$_2$O$_2$ to H$_2$O and O$_2$ (Fig. 4a)[56–58]. To assess the CAT-like activity of FeN$_4$O$_2$-SACs, changes in the UV absorbance for H$_2$O$_2$ at 240 nm were measured initially. Fig. 4d, e shows clear time- and concentration-dependent decreases of the absorbance of H$_2$O$_2$ at fixed concentration of FeN$_4$O$_2$-SACs but varied concentrations of H$_2$O$_2$, or vice versa. A similar result is obtained when FeN$_4$O$_2$-SACs are replaced by natural CAT (Supplementary Fig. 15). We next quantified the elimination rate of H$_2$O$_2$. From Supplementary Fig. 16, the elimination rate of H$_2$O$_2$ by 0.54 μg/ml Fe in FeN$_4$O$_2$-SACs is comparable to that of 62.5 μg/ml natural CAT. Additionally, the Michaelis constant ($K_m$) and maximum reaction velocity ($V_m$) are calculated from Lineweaver–Burk plots (Fig. 4f). The $K_m$ values of FeN$_4$O$_2$-SACs and natural CAT are 9.16 and 45.906 mM, respectively, and the $V_m$ values are 5.856 and 20.533 mM/min, respectively, which suggest that the natural CAT exhibits higher catalytic activity, whereas FeN$_4$O$_2$-SACs possess stronger affinity to H$_2$O$_2$. Subsequently, the effects of pH on FeN$_4$O$_2$-SACs and natural CAT activity were scrutinized. Supplementary Fig. 17 demonstrates that the enzymatic activities of both FeN$_4$O$_2$-SACs and natural CAT increase with increasing pH. More interestingly, FeN$_4$O$_2$-SACs show higher activity than natural CAT in the presence of excess base (pH > 9) because H$_2$O$_2$/O$_2$ has a relatively low redox potential under alkaline condition and is more readily oxidized by Fe$^{3+}$(ref. [59]). Then, the CAT-like performance of FeN$_4$O$_2$-SACs was compared with those of CeO$_2$ and Mn$_3$O$_4$. At the same mass concentration, FeN$_4$O$_2$-SACs markedly outperform the others in promoting the decomposition of H$_2$O$_2$ and producing transparent bubbles (Supplementary Fig. 18). The enzyme activity was further quantified with a CAT assay kit. It is apparent from Fig. 4g that FeN$_4$O$_2$-SACs present the best CAT-like activity, followed by Mn$_3$O$_4$ and CeO$_2$. The calculated enzyme activity of Fe in FeN$_4$O$_2$-SACs is respectively 2121 and 471 times that of Ce in CeO$_2$ and Mn in Mn$_3$O$_4$. Finally, the catalytic activity of other commercially available single-atom catalysts in the decomposition of H$_2$O$_2$ was tested, further confirming the highest catalytic performance of FeN$_4$O$_2$-SACs in comparison with several reported nanozymes (Fig. 4g). Collectively, the above results demonstrate that FeN$_4$O$_2$-SACs possess excellent CAT-like enzymatic activity.

In addition, natural APX also catalyses the dissociation of H$_2$O$_2$ into H$_2$O using ascorbic acid (AsA) as the particular electron donor (Fig. 4a). Here, the APX-like activity was characterized by monitoring the absorbance of AsA at 290 nm. As Fig. 4h, i illustrates, FeN$_4$O$_2$-SACs

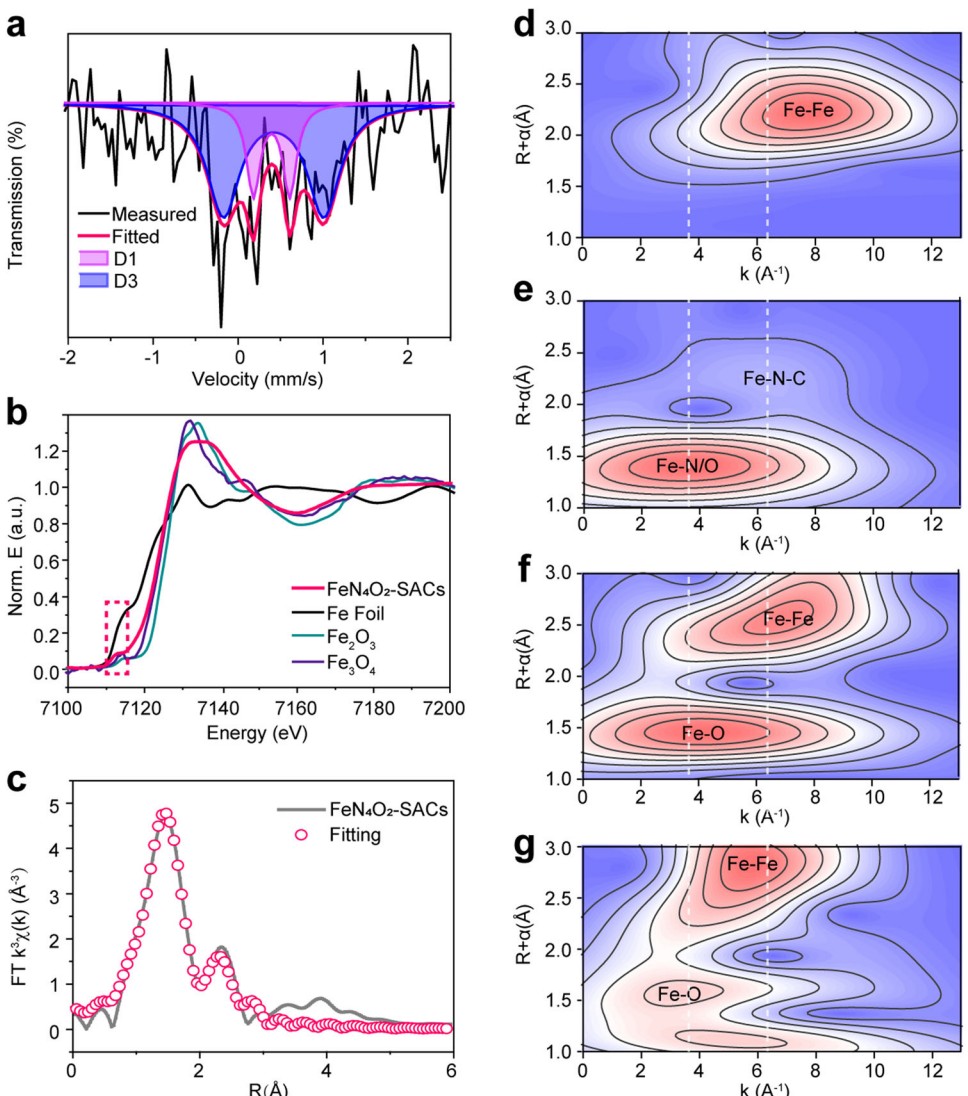

**Fig. 3 | Structure of FeN₄O₂-SACs. a**[57] Fe Mössbauer spectrum. **b** Normalized XANES spectra of FeN₄O₂-SACs and reference samples (Fe Foil, Fe₂O₃, Fe₃O₄). **c** EXAFS fitting in R space. **d–g** Wavelet transformation of Fe Foil (**d**), FeN₄O₂-SACs (**e**), Fe₂O₃ (**f**), and Fe₃O₄ (**g**). Source data are provided as a Source Data file.

exhibit a much higher activity in promoting the elimination of AsA than Co-SACs and Cu-SACs in the presence of $H_2O_2$, while $Mn_3O_4$, $CeO_2$ and Zn-SACs hardly exhibit such activities. The calculated AsA elimination activities of various materials are listed in Supplementary Table 3.

In summary, we have demonstrated that FeN₄O₂-SACs manifest a homogeneous, amorphous, dodecahedral structure with single Fe-N₄O₂ active sites being embedded in the structure, which endows the materials with remarkable or significant SOD-, CAT-, and APX-like activities. Such activities are greatly higher than those of typical anti-inflammatory nanozymes ($Mn_3O_4$ and $CeO_2$) and commercial Co-SACs, Cu-SACs, and Zn-SACs, even better than natural enzymes under some reaction conditions. In addition, compared with other antioxidant nanozymes of SACs reported in the literature (Supplementary Table 4), such as single-atom Ir enzyme mimics (Ir NC SAzymes)[58] and single-atom cobalt nanozymes (Co-SAzymes)[59], FeN₄O₂-SACs exhibit superior catalytic activities. It can be inferred that the multienzyme activity is related to the fact that FeN₄O₂-SACs have a structure analogous to those of natural anti-inflammatory enzymes.

## Hyperproliferation and inflammatory infiltration inhibition of HaCaT cells by FeN₄O₂-SACs via ROS scavenging

To investigate the impact of FeN₄O₂-SACs on HaCaT cells, different concentrations of FeN₄O₂-SACs were incubated with the cells. It is

found that the cell viability decreases with the increasing FeN₄O₂-SACs concentration. The IC₅₀ of FeN₄O₂-SACs on HaCaT cells is determined to be 87.8 µg/ml (Supplementary Fig. 19). Additionally, the cytotoxicity profile of FeN₄O₂-SACs on normal human epidermal keratinocytes (NHEK cells) was also evaluated, and its antiproliferation and anti-inflammation effects can be clearly seen (Supplementary Figs. 20, 21).

Since the oxidative damage induced by ROS contributes to the development of psoriasis, the effect of FeN₄O₂-SACs on total cellular ROS levels was first detected by flow cytometry (Fig. 5a–c and Supplementary Fig. 22) and fluorescence microscope (Supplementary Fig. 23), which was based on the knowledge that ROS can oxidize nonfluorescent DCFH to generate fluorescent DCF. In untreated HaCaT cells, FeN₄O₂-SACs addition can hardly affect the normal ROS level. Interestingly, in contrast to the overactive ROS production induced by M5 (a composition of five proinflammatory factors containing IL-22, IL-17A, IL-1α, TNF-α, and oncostatin M) in HaCaT cells, there is a great decrease in ROS accumulation in the M5 + FeN₄O₂-SACs group. Similarly, reductions in ROS are observed in NHEKs by flow cytometry (Supplementary Fig. 24), which is thought to be advantageous to psoriasis therapy.

To further determine whether FeN₄O₂-SACs could function in psoriasis treatment or not in vitro, the upregulated proliferation of HaCaT cells and inflammation were first established in M5-treated

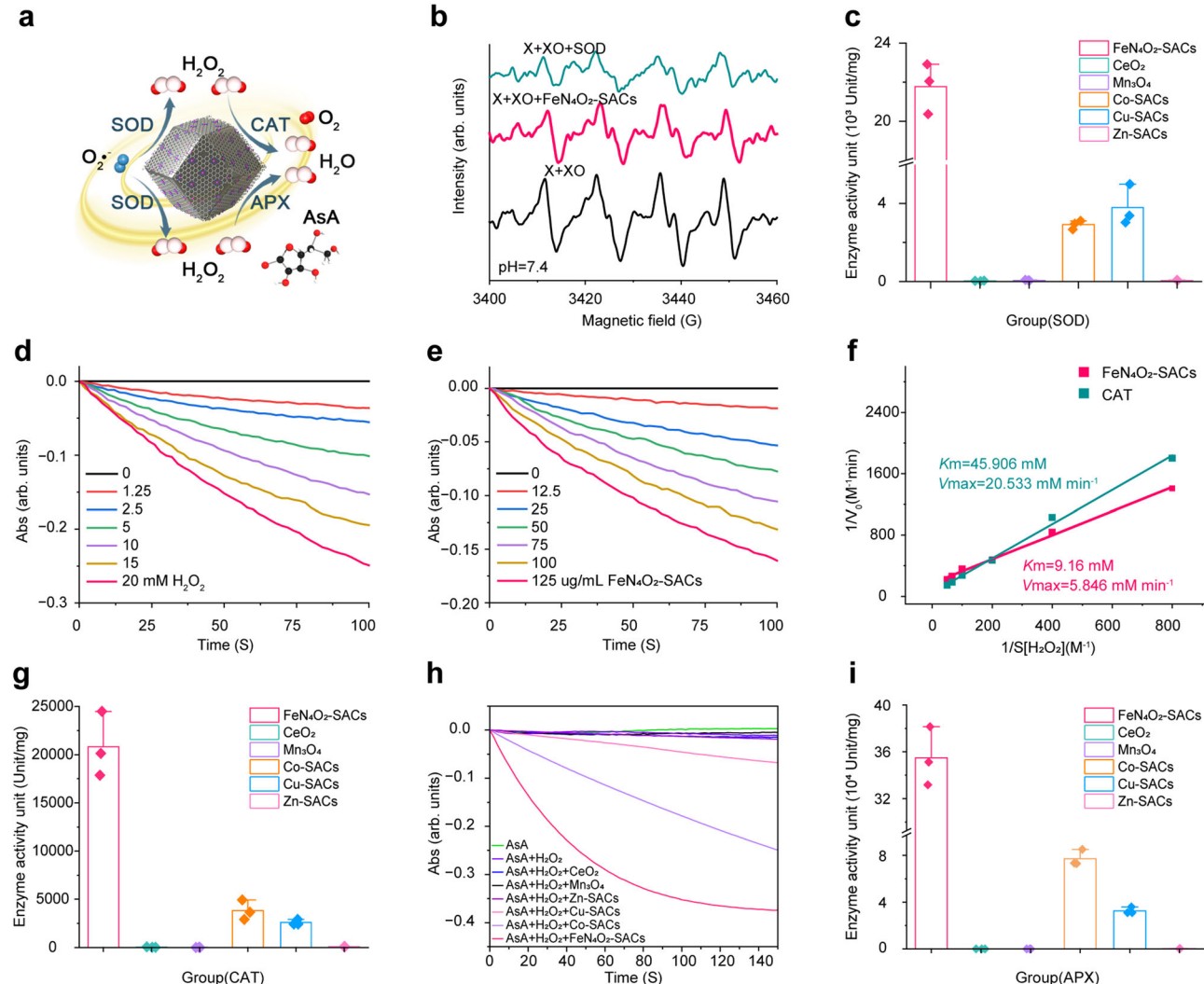

**Fig. 4 | CAT, SOD, and APX-like activities of FeN₄O₂-SACs. a** Schematic illustration of ROS clearance by FeN₄O₂-SACs. **b** ESR spectra of FeN₄O₂-SACs and natural SOD for O₂• clearance at pH=7.4 (X: xanthine; XO: xanthine oxidase). **c** Comparison of SOD-like activities between FeN₄O₂-SACs and control materials CeO₂, Mn₃O₄, Co-SACs, Cu-SACs and Zn-SACs by WST-8 colorimetric analysis. $n = 3$ independent experiments and data are presented as mean ± SD. **d** CAT-like assay for eliminating varied concentrations of H₂O₂ using 125 µg/ml FeN₄O₂-SACs by UV absorption tests. **e** CAT-like assay for eliminating 10 mM H₂O₂ using FeN₄O₂-SACs of varied concentrations. **f** Lineweaver–Burk plots for FeN₄O₂-SACs and natural CAT at varied

concentrations of H₂O₂ (0-20 mM). **g** Comparison of the CAT-like activities among FeN₄O₂-SACs and control materials Mn₃O₄, CeO₂, Co-SACs, Cu-SACs and Zn-SACs. $n = 3$ independent experiments and data are presented as mean ± SD. **h** AsA characteristic absorption intensity decreases upon the additions of H₂O₂ of different samples. **i** Comparison of the APX-like activities between FeN₄O₂-SACs and other materials Mn₃O₄ and CeO₂, Co-SACs, Cu-SACs and Zn-SACs. $n = 3$ independent experiments and data are presented as mean ± SD. Source data are provided as a Source Data file.

HaCaT cells, followed by FeN₄O₂-SACs treatment. Attractively, the FeN₄O₂-SACs treatment has largely weakened the M5-stimulated psoriasis-like morbidity status and shows inhibited cell proliferation and reduced *TNF-α*, *IL-6*, and *IL-8* mRNA expressions (Fig. 5d, e). It is suggest that FeN₄O₂-SACs can remarkably rescue the hyperproliferation and excessive inflammation of psoriasis.

**Alleviation of skin lesions in IMQ-induced psoriasis-like dermatitis mice**

To confirm the therapeutic effects of FeN₄O₂-SACs on psoriasis lesions, we utilized imiquimod (IMQ) cream to establish psoriasis-like mouse models and then treated them with FeN₄O₂-SACs (Fig. 6a). FeN₄O₂-SACs at different concentrations (1.25 µg/ml, 2.5 µg/ml, 5 µg/ml) consistently reduce the ear thicknesses, psoriasis area and severity index (PASI) score of mouse lesions, among which 2.5 µg/ml FeN₄O₂-SACs showed the most prominent effects and were used in follow-up experiments (Supplementary Fig. 25). As shown in Fig. 6b,

the erythematosquamous plaques induced by IMQ have been more effectively alleviated by FeN₄O₂-SACs treatment than by calcipotriol (Cal), and the ear thickness and PASI score are also decreased by the FeN₄O₂-SACs treatment (Fig. 6c, d). Microcosmically, FeN₄O₂-SACs relieve IMQ-induced epidermal hyperplasia, acanthosis, and hyperkeratosis with performance in a comparable manner to that of Cal treatment (Fig. 6e, f). Immunohistochemistry staining (IHC) shows that at the lesional skin, the increased numbers of CD3⁺ T cells, F4/80⁺ macrophages, and PCNA⁺ keratinocytes, as well as p-STAT1⁺, p-STAT3⁺ and NF-κB p50⁺ signals[60] developed in the IMQ group have been strikingly diminished in the FeN₄O₂-SACs-treated and Cal groups, and the FeN₄O₂-SACs group exhibits a more significant reduction in p-STAT1⁺, p-STAT3⁺, NF-κB p50⁺ signals, CD3⁺ T cells and F4/80⁺ macrophages than the Cal group (Fig. 6g, h). With FeN₄O₂-SACs treatment, the signal of tissue-resident memory T cells (indicated by CD103⁺) activated by IMQ induction is weakened. Consistently, the flow cytometry results shows that FeN₄O₂-SACs remarkably reduce the

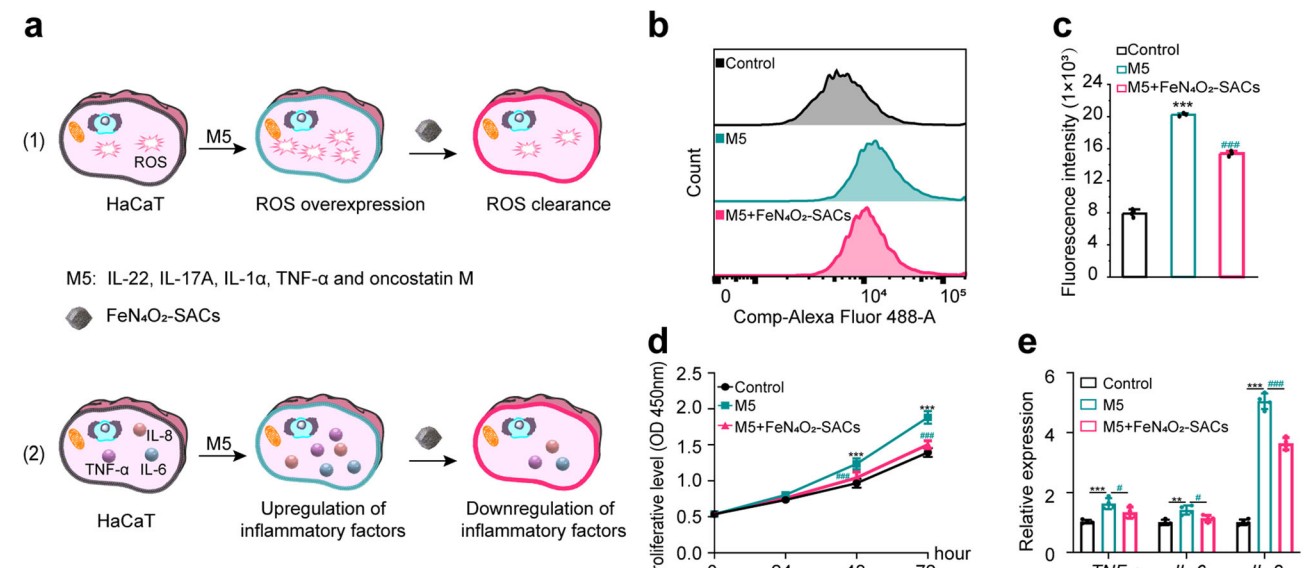

**Fig. 5 | FeN₄O₂-SACs effectively inhibit hyperproliferation and inflammation by scavenging ROS in vitro. a** Schematic illustration of the cell experimental design and results. **b, c** Flow cytometry analysis of HaCaT cells in different groups stained by DCFH-DA (2,7-Dichlorodihydrofluorescein diacetate, EX498/EM530 nm) (**b**) and the quantification of fluorescence intensity (**c**). $n = 3$ independent experiments and data are presented as mean ± SD. **d** CCK-8 assay showing the cell proliferative viabilities of different groups. $n = 3$ independent experiments and data are

presented as mean ± SD. **e** RT-qPCR results for the inflammatory factors *TNF-α*, *IL-6*, and *IL-8* at 24 hours. $n = 4$ independent experiments and data are presented as mean ± SD. ***$p < 0.001$, **$p < 0.01$ versus the Control group; ###$p < 0.001$, #$p < 0.05$ versus the M5 group. Statistical significance was calculated via one-way ANOVA (**c**), two-way ANOVA (**d**), and two independent samples unpaired Student's *t* test (**e**). Source data are provided as a Source Data file.

CD3⁺T cells, F4/80⁺CD11b⁺macrophages and CD103⁺CD8⁺T_RM cells (Supplementary Figs. 26–28)[61]. Also, the mRNA levels of the proinflammatory cytokines *Il-17a*, *Tnf-α*, *Il-12*, and *Il-23* are markedly elevated in the IMQ group but greatly downregulated by FeN₄O₂-SACs treatment, similar to the Cal group (Fig. 6i). The anti-inflammatory factor *Il-10* is downregulated by IMQ induction, and FeN₄O₂-SACs significantly elevate its level (Supplementary Fig. 29). The in vivo ROS level evaluation reveals that FeN₄O₂-SACs treatment significantly ameliorates the oxidative stress induced by IMQ (Fig. 6j, k). Additionally, the organ toxicity of FeN₄O₂-SACs in treating psoriasis-like dermatitis was evaluated and revealed no safety concerns (Supplementary Fig. 30). In conclusion, FeN₄O₂-SACs demonstrate a significant therapeutic effect on IMQ-induced psoriasis-like dermatitis mice by attenuating keratinocyte abnormality, excessive inflammation, and oxidative damage, showing great application potential in clinical psoriasis treatment without side effects.

### Transcriptional profiles and GSEA analysis of FeN₄O₂-SACs-treated mice

RNA sequencing analyses of lesional skins before and after catalytic therapy were utilized to reveal the regulatory mechanism of FeN₄O₂-SACs in treating IMQ-induced psoriasis-like dermatitis (Fig. 7a). As a result, 1546 differentially expressed (DE) mRNAs ($p < 0.05$, |Log₂FoldChange| ≥ 2) are identified, involving 1216 upregulated and 330 downregulated mRNAs (Fig. 7b). To assess the functional enrichment of DE mRNAs, GO annotations and KEGG pathways were evaluated. The top 10 GO terms are found to be: epidermal development, skin development, collagen-containing extracellular matrix, intermediate filament, epidermal cell differentiation, intermediate filament cytoskeleton, external encapsulating structure organization, molting cycle, hair cycle, and extracellular matrix structural constituent (Fig. 7c), meanwhile the top 10 KEGG pathways are identified to be: bAsAl cell carcinoma, Staphylococcus aureus infection, Wnt signaling pathway, Cushing syndrome, cell adhesion molecules, estrogen signaling pathway, melanogenesis, signaling pathways regulating pluripotency of stem cells, amoebiasis and Hippo signaling pathway (Fig. 7d). These

evidences indicate that FeN₄O₂-SACs are able to ameliorate psoriasis-like dermatitis by altering the expressions of various DE mRNAs and functioning in skin/dermal development and inflammatory cascades.

To address the essential role of FeN₄O₂-SACs in regulating ROS, Gene Set Enrichment Analysis (GSEA) was performed to acquire the ROS-signal enrichment curve by including all the genes regulated by FeN₄O₂-SACs. As shown in Fig. 7e, the gene members with greater contribution to the enrichment score are located on the right side of the peak, and the enrichment curve is clustered on the right side. The enrichment score is negative, indicating that compared with the IMQ group, FeN₄O₂-SACs administration downregulate the ROS signal pathway. Additionally, we obtained the ROS-related DE mRNAs from KEGG pathways and analyzed them with a *t* test (Supplementary Fig. 31. It can be found that *Fos*, *Hif1a*, *Nfkbia*, *Nqo1*, and *Slc26a9* are significantly downregulated by FeN₄O₂-SACs compared with the IMQ group (Fig. 7f). Overall, FeN₄O₂-SACs can be applied as an inhibitor of ROS.

### Identification and functional analysis of the key protein ESR1

In order to identify the hub gene of FeN₄O₂-SACs that ameliorates psoriatic dermatitis, the Protein-Protein Interaction (PPI) network and important modules were screened from the PPI network. Cytoscape was used to analyze the PPI networks of the cross-targets obtained from String databases. Through Cytoscape's MCODE plug-in, two closely linked gene modules with degree are identified (module 1, score = 15.879; module 2 (not shown), score = 11.306). A total of 34 targets and 262 interaction pairs are included in module 1, and 50 targets along with 277 interaction pairs are included in module 2. Based on the fact that most of the targets are found in module 1, we performed a topological network analysis to evaluate the regulatory relationship between targets. Topological parameters were used to screen a 40-node, 774-edge subnetwork: BC > Avg (BC), CC > Avg (CC), and De > Avg (De). A kernel network was extracted based on the subnetwork in accordance with De ranking. It consists of 17 nodes and 105 edges, with ESR1 being the core node (Fig. 7g). The *ESR1* gene encodes an estrogen receptor (ESR) related to growth, sexual

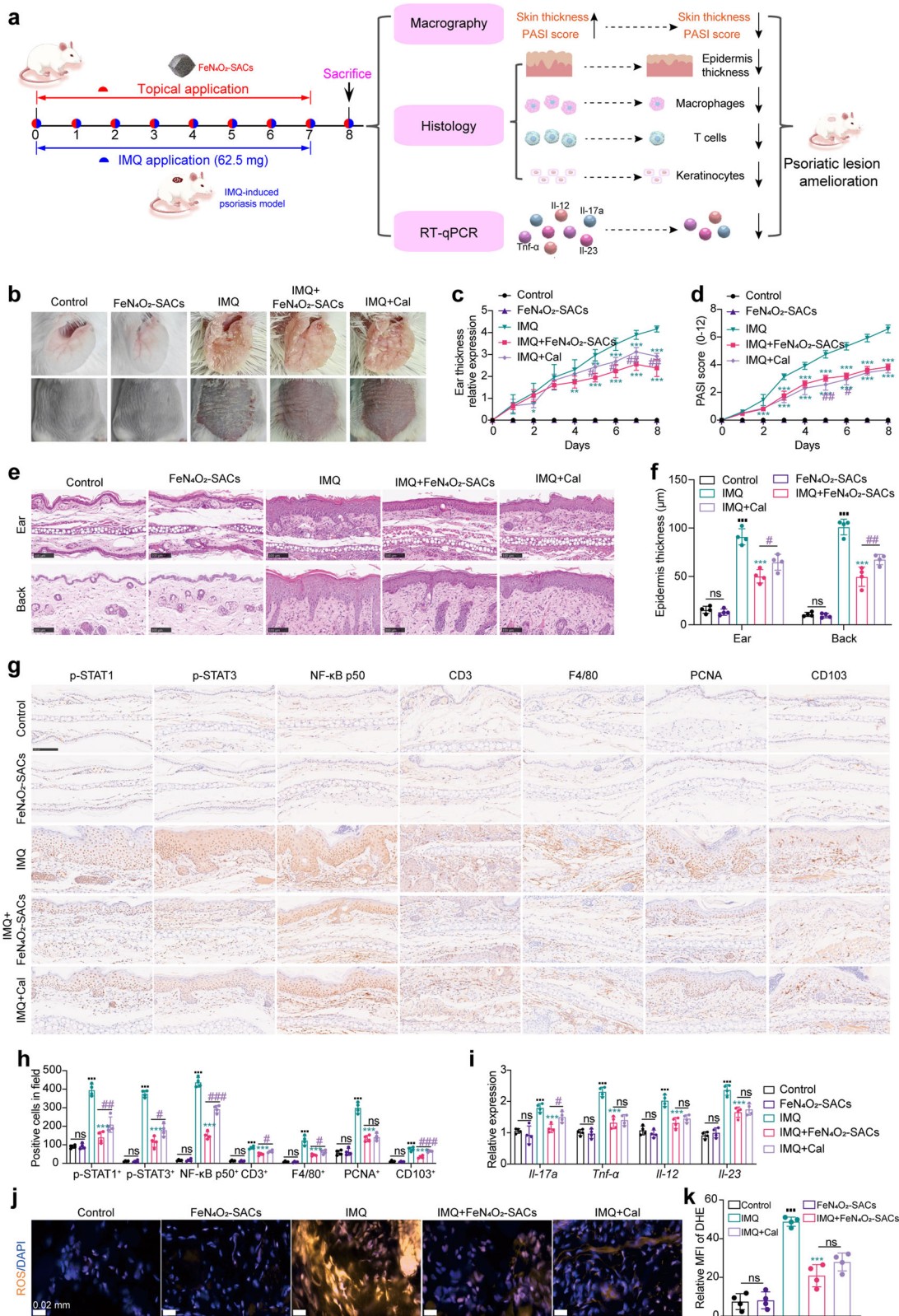

development, and metabolism[62]. ESR1 is significantly reduced in lesions of psoriatic patients[63,64], and psoriatic inflammation exacerbates in mice with neutrophils and macrophages lacking ESR (*Esr1*^f/f *Esr2*^f/f*LysM*-Cre⁺ mice)[65]. Herein, we find that the mRNA and protein levels of ESR1 are downregulated in psoriasiform skin tissues of mice, whereas FeN$_4$O$_2$-SACs rescue the IMQ-induced ESR1 under-expression (Fig. 7h, i).

By silencing ESR1 with shRNA (shESR1, Supplementary Fig. 32), aggravated cell proliferation and inflammation are observed in M5-induced psoriasis-like in vitro model (Fig. 7j, k), and notably, ROS-related hub gene upregulations can be seen (Fig. 7l). Then FeN$_4$O$_2$-SACs treatment downregulates the relative mRNA levels of ROS-related genes to large extents, which alleviated the inflammation. Further, ESR1 expression reductions (M5 + FeN$_4$O$_2$-SACs + shESR1)

**Fig. 6 | FeN$_4$O$_2$-SACs effectively alleviate the psoriasis dermatitis in vivo.**
**a** Schemes of the IMQ-induced psoriasiform model, treatments and efficacy.
**b** Representative images of lesions on day 8, $n = 4$ samples/group.
**c, d** Quantification of ear thickness (**c**) and PASI score (**d**), $n = 4$ samples/group and data are presented as mean ± SD. **e, f** Histological staining (**e**) and the corresponding quantifications (**f**). Scale bar=100 μm, $n = 4$ samples/group and data are presented as mean ± SD. **g, h** Levels of inflammatory signals, T cells, macrophages, proliferative keratinocytes and tissue-resident memory T cells analyzed by immunohistochemistry labelling of p-STAT1, p-STAT3, NF-κB p50, CD3, F4/80, PCNA and CD103 in the ear lesions on day 8 (**g**), and the corresponding quantifications (**h**). Scale bar=100 μm, n = 4 samples/group and data are presented as mean ± SD. **i** The

mRNA expression levels of *Il-17a, Tnf-α, Il-12,* and *Il-23* at the ear skin tissues detected by RT-qPCR. $n = 4$ samples/group and data are presented as mean ± SD. **j, k** The ROS level of ear skins from different groups detected by fluorescence probe DHE (Dihydroethidium, Filter block of spORANGE at EX532-554/EM576-596 nm) (**j**) and the quantification (**k**) of relative mean fluorescence intensity (MFI), orange color indicates the ROS-positive, n = 4 samples/group and data are presented as mean ± SD. $^{***}p < 0.001$, $^{**}p < 0.01$, $^{*}p < 0.05$ versus the IMQ group; $^{###}p < 0.001$, $^{##}p < 0.01$, $^{#}p < 0.05$ versus the IMQ+Cal group; $^{■■■}p < 0.001$ versus the Control group; ns means no significance. Statistical significance was calculated via two-way ANOVA (**c, d**), two independent samples unpaired Student's $t$ test (**f, h, i, k**). Source data are provided as a Source Data file.

again reverse the amelioration effects over hyperproliferation, excessive inflammation and ROS generation by FeN$_4$O$_2$-SAC treatment, resulting in keratinocyte hyperproliferation, inflammatory cytokine overexpression and ROS-associated gene upregulations. Conversely, when the expression of ESR1 was knocked down using CRISPR/Cas9, FeN$_4$O$_2$-SACs treatment is not able to rescue the hyperproliferation and undue inflammation of M5-induced psoriasis-like in vitro model, indicating that ESR1 is indispensable for the curative effects of FeN$_4$O$_2$-SACs (Supplementary Fig. 33). In summary, FeN$_4$O$_2$-SACs are highly effective in psoriasis control, hyperproliferation restraint, inflammation inhibition and ROS reduction by upregulating ESR1.

### Effective prevention of psoriasis relapse by FeN$_4$O$_2$-SACs

Two datasets (GSE52471, and GSE14905) related to psoriatic patients from the GEO database were adopted to evaluate the clinical relevance of ESR1, and 34 normal samples and 51 psoriasis samples were obtained (Supplementary Fig. 34a, b). By analyzing the expression levels of ESR1 in normal and psoriatic patients, we observe that psoriatic individuals exhibit significant reductions in ESR1 compared with normal individuals (Fig. 8a, b and Supplementary Fig. 34c). Notably, the recurrence of psoriasis is a prevalent and intractable problem in psoriasis therapy. IHC results verify that the ESR1 level is lower in psoriatic and recurrent lesions than in normal skin (Fig. 8a, b), indicating the clinical responsibility of ESR1 deficiency in psoriasis pathology and that FeN$_4$O$_2$-SACs may ameliorate psoriasis and avoid relapse by restoring ESR1.

We established a relapse model of psoriasis of the mice with secondary but lower doses of IMQ for 10 days after 8-days of treatment and 30-days of rest (Fig. 8c). During the resting period, the dry and rough skin gradually recovered with time in all groups, while after another 10-days of re-stimulation by IMQ, the mice pre-treated with PBS and Cal showed obvious erythematosquamous plaques, and increased epidermal thickness and inflammatory factor levels (*Il-17a, Tnf-α, Il-12, Il-23*). Intriguingly, the skin of previously FeN$_4$O$_2$-SACs-treated mice exhibits mild psoriasiform lesions and ear thickness compared with the Non-relapse group (the 8-days IMQ-induced psoriasis mice received no secondary IMQ induction with or without FeN$_4$O$_2$-SACs treatment) (Fig. 8d–g and Supplementary Fig. 35). FeN$_4$O$_2$-SACs application has largely ameliorated IMQ-induced epidermal thickening and inflammatory infiltration on day 10 after the second stimulation compared with PBS/Cal. IHC results show that in contrast with the increased inflammatory cells and cell proliferation in the Relapse and Relapse+Cal groups, FeN$_4$O$_2$-SACs treatment significantly diminishes the re-infiltration of p-STAT1$^+$, p-STAT3$^+$, NF-κB p50$^+$ signals, CD3$^+$ T cells, F4/80$^+$ macrophages, and PCNA$^+$ keratinocytes (Fig. 8h and Supplementary Fig. 36). With FeN$_4$O$_2$-SACs, the inflammatory cytokins *Il-17a, Tnf-α, Il-12, Il-23* are inhibited even with IMQ induction (Fig. 8i). The in vivo ROS detection shows that FeN$_4$O$_2$-SACs treatment significantly prevents the mice from the damage of oxidative stress (Supplementary Fig. 37). Additionally, the level of ESR1 is sustained by FeN$_4$O$_2$-SACs compared with the Relapse group (Supplementary Fig. 38). These data suggest that psoriasis catalytic therapy is an effective therapeutic modality for preventing disease rebounding of psoriasis-like skin lesions through ESR1, much better than

conventional Cal treatment, in addition to its satisfactory efficacy in the initial treatments.

## Discussion

In this work, a "psoriasis catalytic therapy" strategy has been introduced to ameliorate psoriatic lesions and prevent its recurrence by designing catalytically active biomimetic materials (FeN$_4$O$_2$-SACs) possessing multiple CAT-, SOD- and APX-like activities to continuously scavenge overexpressed ROS. Notably, the multienzyme-like activities of Fe in FeN$_4$O$_2$-SACs greatly outperform those of Ce in CeO$_2$, Mn in Mn$_3$O$_4$ and Zn in Zn-SACs by several orders of magnitude (Summary in Supplementary Table 3). As an efficient treatment modality, FeN$_4$O$_2$-SACs are capable of effectively and continuously inhibiting hyperproliferation and inflammatory infiltration in psoriasis by ROS elimination. Moreover, the PASI score, epidermal thickness, lymphocyte infiltration, macrophage infiltration, proliferation of keratinocytes and levels of proinflammatory cytokines in IMQ-induced psoriasis-like mouse models can be significantly inhibited by FeN$_4$O$_2$-SACs, demonstrating even better efficacy than that of clinical Cal. Mechanistically, ESR1 deficiency has been identified by RNA sequencing and bioinformatic analysis to play a key role in psoriasis development and relapse, which, attractively, could be inhibited to large extents by FeN$_4$O$_2$-SACs treatment via ESR1 upregulation. More importantly, a relapse model of psoriasis evidences the significant reductions in psoriasis recurrence by FeN$_4$O$_2$-SACs treatment. The present catalytic therapy strategy opens up a promising therapeutic pathway for psoriasis amelioration and relapse prevention, which serves as a highly effective example of multienzyme-inspired bionics.

## Methods
### Materials

Zinc nitrate hexahydrate (98%, Zn(NO$_3$)$_2$·6H$_2$O) and ferric nitrate nonahydrate (98%, Fe(NO$_3$)$_3$·9H$_2$O) were purchased from Alfa Aesar. Methanol and ethanol were purchased from Sinopharm. 2-Methylimidazole (99%), superoxide dismutase (≥2500 units/mg protein), and xanthine oxidase (from milk, ≥0.4 units/mg protein) were obtained from Sigma-Aldrich. PBS was obtained from YoBiBio. Catalase (from bovine liver, 20000 units/mg) was obtained from Adamas. Catalase Assay Kit and Total Superoxide Dismutase Kit were purchased from Shanghai Beyotime Biotechnology Co., Ltd. Ascorbic acid, xanthine (98%), hydrogen peroxide (H$_2$O$_2$, 30 wt%), and sulfuric acid (98%) were purchased from Aladdin Co. Ltd. 5-tert-butoxycarbonyl 5-methyl-1-pyrroline N-oxide (BMPO) were purchased from Shanghai Dojindo Co. Ltd.

### SOD-like activity

The O$_2^{•-}$ clearance ability of FeN$_4$O$_2$-SACs and natural SOD were verified by ESR measurements. Typically, 31.25 μg/ml Fe-SAs/NC or 0.49 μg/ml natural SOD was added to a mixture of 25 mM BMPO, 1 mM xanthine, and 0.0125 U/mL xanthine oxidase in different pH PBS solutions (pH 6, 7.4, 8).

The SOD-like activity of FeN$_4$O$_2$-SACs, NC, CeO$_2$, and Mn$_3$O$_4$ was further quantified using a SOD assay kit (WST-8) according to the

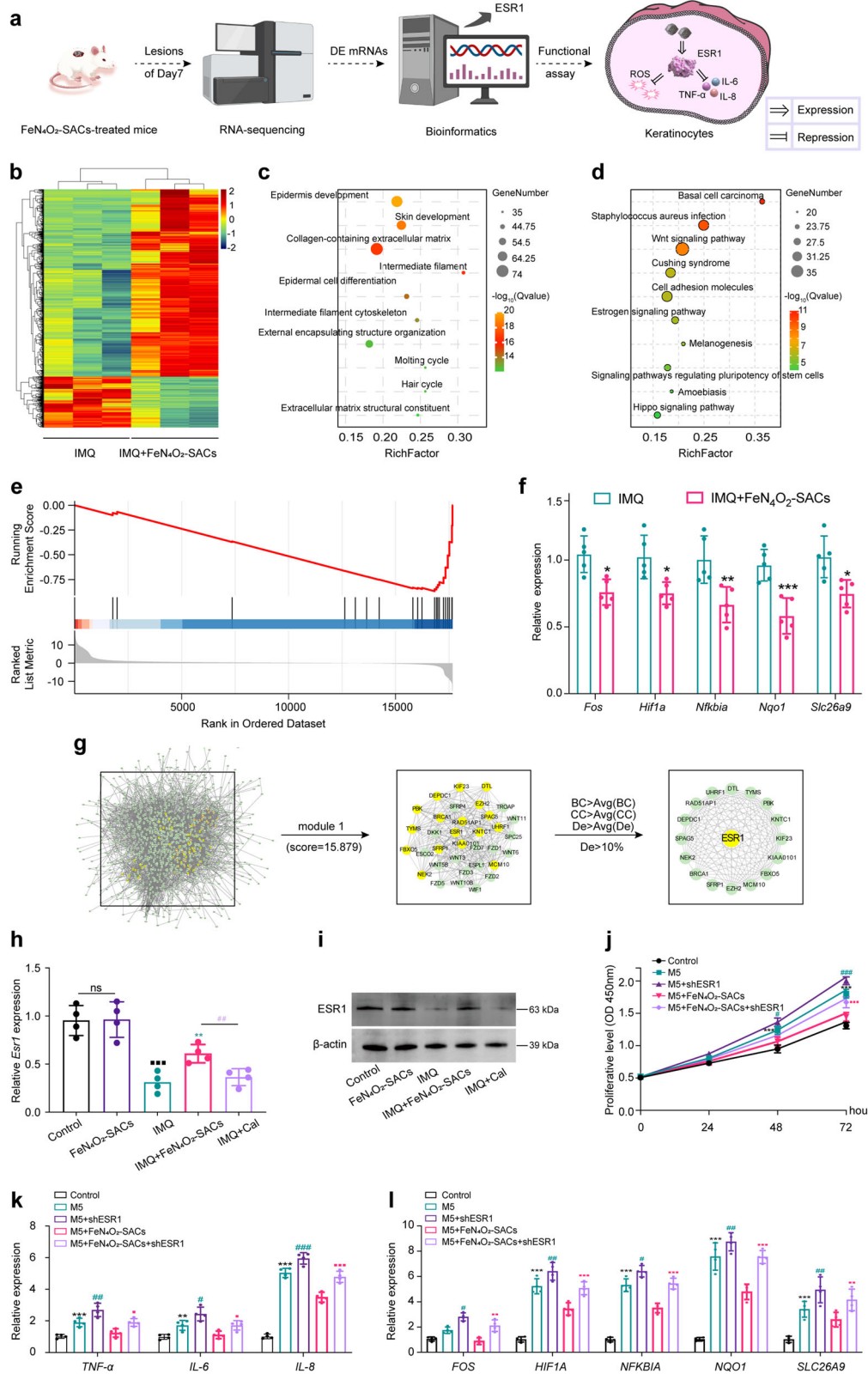

manufacturer's instructions. Specifically, $O_2^{\bullet-}$ is commonly generated through the action of xanthine and xanthine oxidase, and further reacts with WST-8 to form the formazan dye with a characteristic absorption at 450 nm. Hence, the inhibition rate of $O_2^{\bullet-}$ is available by colorimetric analysis of the WST-8 product. Correspondingly, various samples with concentrations of 15.625, 31.25, and 62.5 µg/ml were added to an equal volume of detection reagent. After approximately

30 min of incubation, the characteristic absorption change at 450 nm was detected using SpectraMax M2 Molecular Devices. The inhibition percentage was determined based on the following equation: inhibition (%) = $[(A_0-A)/A_0] \times 100\%$, where A is the absorbance of various samples and $A_0$ is the absorbance of the control. The inhibition rate of natural SOD (0.12–3.9 µg/ml) against superoxide anion was assayed using the same method.

**Fig. 7 | RNA-sequencing of mouse skin lesions and the identification of key protein ESR1. a** Schematic illustration of the underlying mechanism analysis of FeN$_4$O$_2$-SACs treatment. **b** The heatmap of differentially expressed (DE) mRNAs ($n$ = 3). **c, d** GO annotation (**c**) and KEGG pathway enriched (**d**) by DE mRNAs. **e** The enrichment curve of ROS signal (FeN$_4$O$_2$-SACs versus IMQ). **f** The core genes downregulated in the ear lesions of IMQ+FeN$_4$O$_2$-SACs group validated by RT-qPCR, $n$ = 5 samples/group and data are present as mean ± SD. $^{***}p < 0.001$, $^{**}p < 0.01$, $^{*}p < 0.05$ versus the IMQ group. **g** Protein network of the DE mRNAs. **h** RT-qPCR validation of *Esr1* in ear skin tissues, $n$ = 4 samples/group and data are present as mean ± SD. $^{**}p < 0.01$ versus the IMQ group; $^{##}p < 0.01$ versus the IMQ+FeN$_4$O$_2$-SACs group; $^{\blacksquare\blacksquare\blacksquare}p < 0.001$ versus the Control group; ns means no significance. **i** Western blotting analysis of ESR1 protein in the ear lesions of mice from different groups. The samples derive from the same experiment and that gels/blots were processed in parallel. **j** CCK-8 assay detecting the cell proliferative activities of different groups. $n$ = 3 independent experiments and data are present as mean ± SD. **k** The mRNA expression levels of inflammatory factors *TNF-α*, *IL-6*, and *IL-8* detected by RT-qPCR at 24 hours. $n$ = 4 independent experiments and data are present as mean ± SD. **l** Relative mRNA levels of ROS-related genes determined by RT-qPCR at 24 hours. $n$ = 4 independent experiments and data are present as mean ± SD. $^{***}p < 0.001$, $^{**}p < 0.01$, versus the Control group; $^{###}p < 0.001$, $^{##}p < 0.01$, $^{#}p < 0.05$ versus the M5 group; $^{\blacksquare\blacksquare\blacksquare}p < 0.001$, $^{\blacksquare\blacksquare}p < 0.01$, $^{\blacksquare}p < 0.05$ versus the M5 + FeN$_4$O$_2$-SACs group. Statistical significance was calculated via two-way ANOVA (**j**), two independent samples unpaired Student's t test (**f, h, k, l**). Source data are provided as a Source Data file.

## CAT-like activity

The CAT-like activity assays of FeN$_4$O$_2$-SACs and natural CAT were verified by measuring the decline in absorbance of H$_2$O$_2$ at 240 nm (39.4 M$^{-1}$ cm$^{-1}$) on a UV-Vis spectrophotometer operating under kinetic mode[66]. Typically, experiments were carried out in PBS buffer solution (pH range from 3 to 11) with a total volume of 2.5 ml containing FeN$_4$O$_2$-SACs (125 µg/ml) or natural CAT (62.5 µg/ml) and H$_2$O$_2$ (10 mM), and the absorbance was collected over time. Meanwhile, kinetic analysis was carried out by varying the concentrations of FeN$_4$O$_2$-SACs (0–125 µg/ml) or natural CAT (0–75 µg/ml) at fixed concentrations of H$_2$O$_2$ (10 mM) in PBS solution and by varying the concentrations of H$_2$O$_2$ (0–20 mM) at fixed concentrations of FeN$_4$O$_2$-SACs (125 µg/ml) or natural CAT (62.5 µg/ml) under identical conditions.

The CAT-like activity of FeN$_4$O$_2$-SACs, NC, CeO$_2$, and Mn$_3$O$_4$ was further determined by a Catalase Assay Kit, and the OD520 was recorded by SpectraMax M2 Molecular Devices. A digital picture of H$_2$O$_2$ elimination was also captured.

## APX-like activity

The ascorbate peroxidase (APX)-like activity of FeN$_4$O$_2$-SACs, NC, CeO$_2$, and Mn$_3$O$_4$ was verified by monitoring the absorption of ascorbate with H$_2$O$_2$ by UV–Vis spectroscopy at 290 nm (2800 M$^{-1}$ cm$^{-1}$). The spectra were collected in PBS (pH=7.4) with a total volume of 2.5 ml containing 0.2 mM ascorbate, 0.02 mg/ml various samples, and H$_2$O$_2$ (10 mM).

## Cell culture and treatment

The human immortal keratinocyte line HaCaT (Cell Lines Service, Eppelheim, 300493) was cultured in DMEM (HG) supplemented with 10% FBS, 100 µg/ml streptomycin, and 100 U/ml penicillin at 37 °C with 5% CO$_2$. Normal human epidermal keratinocytes (NHEK cells, No. 340593) were acquired from Bena Culture Collection (Henan, China) and cultured with CM8-1 culture medium that contains 90% EMEM and 10% FBS. For treatments, different concentrations of FeN$_4$O$_2$-SACs in PBS (2.5 µg/ml, 5 µg/ml, 10 µg/ml, 20 µg/ml, 40 µg/ml, 80 µg/ml, 160 µg/ml, 320 µg/ml) were assayed and 2.5 µg/ml was determined for the subsequent experiments in vitro. For in vitro models of psoriasis, a mixture containing IL-22, IL-17A, IL-1α, TNF-α, and oncostatin M (M5 cocktail, each of 10 ng/ml) was utilized[67,68].

## In vitro ROS detection

ROS assay Kit (50101ES01, Yeasen) was utilized to detect the ROS level in HaCaT cells according to the manufacturer's instruction. In Brief, cells in different groups were treated with Control, M5, and M5 + FeN$_4$O$_2$-SACs for 48 h, respectively. DCFH-DA was diluted with serum-free medium and DCFH-DA was added to the cell suspension and incubated for 30 min at 37 °C in the dark. The fluorescence images were visualized and recorded using a fluorescence microscope at 498/530 nm (excitation/emission) and ×200 magnification.

In flow cytometry analysis, propidium iodide solution (Biolegend, 421301) was added and incubated for 5 min to dye the dead cells. The ROS levels in normal human epidermal keratinocytes (NHEK cells) were also measured using the same procedure.

## In vivo ROS detection

The in vivo ROS level was analyzed using BBoxiProbe® DHE ROS detection kit (BB-47051). According to the manufacturer's instructions, the fluorescence to determine the ROS level in cells was detected by fluorescence microscope at the excitation wavelength of 532–544 nm and the emission wavelength of 576–596 nm. The filter block we used was spOrange, and the intensity of orange fluorescence is proportional to the level of ROS.

## Animals and treatment

Male BALB/c mice aged 6–8 w were provided by Shanghai SLAC Laboratory Animal Co., Ltd., (no. 20220004020279, SYXK (Hu) 2018-0040) and housed under the conditions of 12 h dark/light cycle, an ambient temperature of 22–25 °C and the humidity of 30–70%. Imiquimod (IMQ) cream (H20030129, Sichuan Med-Shine Pharmaceutical Co.,Ltd) was used for psoriasis-like mouse model construction and all therapeutic strategies were performed for 8 days. All mice except those in the control group were topically administered 62.5 mg/mouse IMQ cream on the ears and backs once daily, and then different treatments involving PBS, FeN$_4$O$_2$-SACs or calcipotriol were administered to the same area 6 h later. For determination of FeN$_4$O$_2$-SACs consistency, mice were randomly divided into four intervention groups (62.5 mg/mouse IMQ cream and treatment 6 h later): Control group (PBS, 80 µl/mouse), 1.25 µg/ml group (1.25 µg/ml of FeN$_4$O$_2$-SACs, 80 µl/mouse), 2.5 µg/ml group (2.5 µg/ml of FeN$_4$O$_2$-SACs, 80 µl/mouse), and 5 µg/ml group (5 µg/ml of FeN$_4$O$_2$-SACs, 80 µl/mouse). For therapeutic efficacy of FeN$_4$O$_2$-SACs, mice were randomly arranged into five groups: control group (PBS, 80 µl/mouse), FeN$_4$O$_2$-SACs group (2.5 µg/ml FeN$_4$O$_2$-SACs, 80 µl/mouse), IMQ group (PBS 6 h after IMQ, 80 µl/mouse), IMQ+FeN$_4$O$_2$-SACs group (2.5 µg/ml FeN$_4$O$_2$-SACs 6 h after IMQ), and IMQ+Cal group (1 mg/kg calcipotriol 6 h after IMQ). To evaluate the effectiveness of FeN$_4$O$_2$-SACs in decreasing recurrence, psoriasis relapse models were constructed. Mice were recovered for 30 days after 8-day treatment as mentioned above and then rechallenged with reduced doses of IMQ (20.8 mg/mouse) at the same lesion location for another 10 consecutive days. During the intervention period, the ear thickness of all mice was measured by vernier caliper, and the PASI score of lesion skins was calculated. Mice were executed by CO$_2$ inhalation suffocation at the indicated time-point, and the lesion tissues were collected for further analysis (histology, RT-qPCR, flow cytometry, western blotting, ROS detection, and RNA-seq). All the animal procedures were approved by Ethics Committee of Yueyang Hospital affiliated to Shanghai University of Traditional Chinese Medicine (no. YYLAC-2021-107-6, YYLAC-2022-160-3).

## RNA sequencing

IMQ-induced psoriasis-like mice treated with or without FeN$_4$O$_2$-SACs were executed on day 8 and the lesional skin tissues were lysed with TRIzol to isolate the total RNA. mRNA selection, fragmentation, cDNA

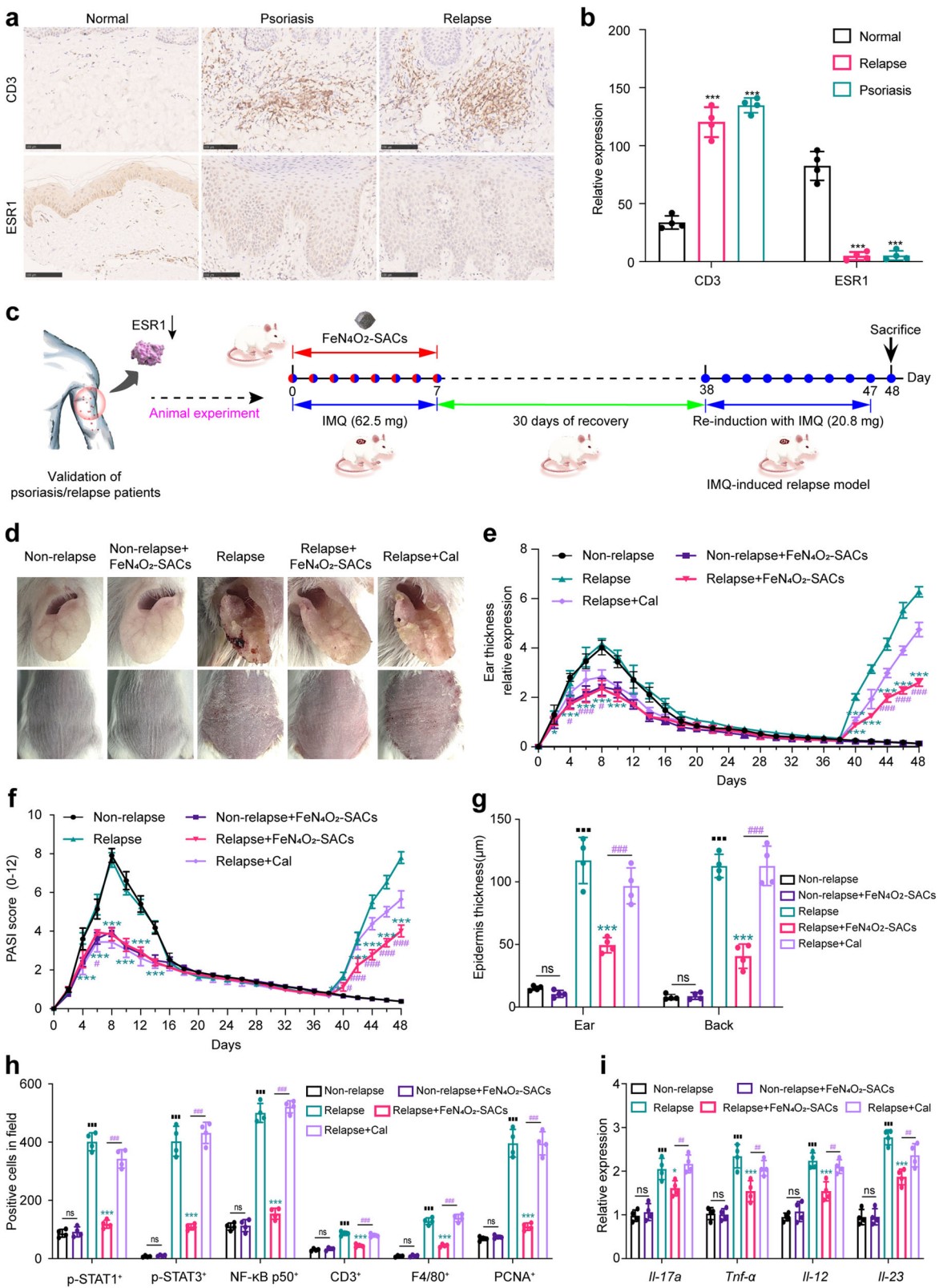

synthesis, and library preparation were performed for each sample, and the TruSeq stranded mRNA LT sample preparation kit (RS-122-2103, Illumina) was used for sequencing. RNA sequencing analysis was conducted by Shanghai Biochip Co., Ltd.. After DE mRNAs screening ($p$ value < 0.05, |Log2Fold-change | ≥ 1), the Gene Ontology (GO) enrichment analysis was carried out through Fisher's test, and the significant enrichment pathways of these DE mRNAs were determined

through the Kyoto Encyclopedia of Genes and Genomes database (KEGG). Additionally, to uncover a more comprehensive biological pathways, the R-package fgsea was employed to conduct the Gene Set Enrichment Analysis (GSEA) on the entire set of genes. This package incorporated both the preranked gene list, obtained from the Molecular Signature Database. The fgsea result was subjected to filtration based on the criterion that the pathway's adjusted $p$-value was less

**Fig. 8 | FeN₄O₂-SACs suppress the recurrence of psoriasis-like skin dermatitis.**
**a** IHC stainings of CD3 and ESR1 in skin tissues of normal, psoriatic, and recurrent individuals. Scale bar = 100 μm. **b** Relative quantifications of CD3 and ESR1 determined by ImageJ. $n = 4$ samples/group and data are presented as mean ± SD. $^{***}p < 0.001$ versus the Normal group. **c** Experimental design of IMQ-stimulated psoriasis relapse model. **d** Representative images of relapse models, $n = 4$ samples/group. **e**, **f** Quantifications of ear thickness (**e**) and PASI score (**f**), $n = 4$ samples/group and data are presented as mean ± SD. **g** Relative quantifications of histological staining. $n = 4$ samples/group and data are presented as mean ± SD. **h**, Quantifications of p-STAT1⁺, p-STAT3⁺, NF-κB p50⁺ signals, CD3⁺ T cells, F4/80⁺

macrophages, and PCNA⁺ keratinocytes in ear skin lesion tissues from different groups. $n = 4$ samples/group and data are presented as mean ± SD. **i** The mRNA expression levels of *Il-17a*, *Tnf-α*, *Il-12* and *Il-23* of ear skin tissues detected by RT-qPCR. $n = 4$ samples/group and data are presented as mean ± SD. $^{***}p < 0.001$, $^{**}p < 0.01$, $^{*}p < 0.05$ versus the Relapse group; $^{###}p < 0.001$, $^{##}p < 0.01$, $^{#}p < 0.05$ versus the Relapse+Cal group; ▀▀▀$p < 0.001$ versus the Non-relapse group; ns means no significance. Statistical significance was calculated via two-way ANOVA (**e**, **f**), two independent samples unpaired Student's t test (**b**, **g**–**i**). Source data are provided as a Source Data file.

than 0.05. The visualization of the KEGG and GSEA pathways was conducted using ggplot2. OE Biotech Co., Ltd (Shanghai, China) provided technical supports for bioinformatic analyses. The raw data of RNA-sequencing has been deposited in the Entrez Molecular Sequence Database System under BioProject accession number of PRJNA909970 (https://www.ncbi.nlm.nih.gov/bioproject/PRJNA909970/).

### Clinical sample
For clinical relevance evaluation of ESR1, healthy individuals and psoriasis patients at both incipient and relapse stages were included with the approval of the Ethics Committee of Yueyang Hospital affiliated to Shanghai University of Traditional Chinese Medicine (no. 2019-29). Skin samples were collected and used for immunohistochemistry (IHC) staining. Written informed consent was obtained from all participants.

### Statistical analysis
Filter and standardize the data before statistical analysis. All values in this study are expressed as mean ± SD. We used Students' *t*-test to analyze the differences between two groups, one-way ANOVA for multiple comparisons and two-way ANOVA for variables repeated at different time points. Statistics were analyzed and displayed by Graphpad Prism 8. The significant differences were determined by $^{***}p < 0.001$, $^{**}p < 0.01$, $^{*}p < 0.05$; $^{###}p < 0.001$, $^{##}p < 0.01$, $^{#}p < 0.05$; ▀▀▀$p < 0.001$, ▀▀$p < 0.01$, ▀$p < 0.05$.

Other experimental methods are detailed in Supplementary Information.

### Names and symbols of genes and proteins
*ESR1*, human gene; *Esr1*, mice gene; ESR1, protein coded by gene[69,70].

### Reporting summary
Further information on research design is available in the Nature Portfolio Reporting Summary linked to this article.

## Data availability
All the data supporting the findings of this study are available within the article, source data, and its supplementary information files. Source data are provided as a Source Data file. The raw data of RNA-sequencing has been deposited in the Entrez Molecular Sequence Database System under BioProject accession number of PRJNA909970. This study uses publicly available data from the Protein Data Bank (PDB) under accession codes: 1DGF, 1V0H, and 1AVM. Source data are provided with this paper.

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

## Acknowledgements

This work was supported by the National Key Research and Development Program of China (Grant No. 2022YFB3804500, 2018YFC1705305), the National Natural Science Foundation of China (Grant No. 22335006, 52202352, 32271387, 82174383, 82374458), the CAMS Innovation Fund for Medical Sciences (Grant No. 2021-I2M-5-012), the Dermatology Department of Traditional Chinese Medicine, the Fundamental Research Funds for the Central Universities (Grant No. 22120230237, 22120220618, 2023-3-YB-11, 2022-4-YB-09), the Natural Science Foundation of Shanghai (Grant No. 23ZR1465200), the Shanghai Municipal Health Commission (Grant No. 20224Y0010), the Youth Talent Promotion Project of China Association of Traditional Chinese Medicine (2021-2023) Category A (Grant No. CACM-2021-QNRC2-A10), the Xinglin Youth Scholar of Shanghai University of Traditional Chinese Medicine (Grant No. RY411.33.10), the "Chen Guang" project supported by Shanghai Municipal Education Commission and Shanghai Education Development Foundation (Grant No. 22CGA50), the Health Young Talents of Shanghai Municipal Health Commission (Grant No. 2022YQ026), the Shanghai Dermatology Research Center (Grant No. 2023ZZ02017), the Shanghai Dermatology Hospital demonstration research ward project (Grant No. SHDC2023CRW009), and the Shanghai Science and Technology Committee (Grant No. 21Y21920101, 21Y21920102). The authors thank beamline BL14W1 (Shanghai Synchrotron Radiation Facility) for providing the beam time.

## Author contributions

J.Shi., X.L., B.L., L.K. and H.D. conceived and designed the study. X.L. and F.H. designed and synthesized the materials, performed the structural characterizations and catalytic experiments, and analyzed the data. L.K. and J.J. performed the in vitro and in vivo study, and analyzed the data. J.Song and Y.Luo assisted with the in vitro and in vivo study. S.C., L.M. and W.P. assisted with the catalytic experiments. Y.Li and Y.Liu provided good advice for the study. J.Shi., X.L., B.L., L.K., H.D., F.H. and J.J. analyzed the results and wrote the paper. All the authors discussed the results and commented on the manuscript.

## Competing interests

The authors declare no competing interests.
