## [Peer Review File · Nature Communications]

Reviewers' Comments:

Reviewer #1:

Remarks to the Author:

Nanozymes are the functional nanomaterials with enzyme-like activities. They have attracted great attention recently, as evidenced that nanozymes were listed at Top Ten Emerging Techniques by IUPAS. Nanozymes have been explored for wide applications. Now, the authors developed a very interesting single atom catalyst with ROS scavenging for psoriasis therapy by upregulating ESR1. The catalytic therapy strategy was also proved effective in relapse prevention, which made nanomaterials with enzyme-like activities promising candidates for skin diseases. Therefore, I would suggest the publication after addressing the following concerns.

Depart from CeO₂ and Mn₃O₄ depicted in Fig.4, other single atom catalyst should be provided to exhibit the advantages of FeN₄O₂-SACs if possible.

In the method section, specifically in "Animal model", it is recommended that additional details were provided, including the construction of the model and the technique employed for FeN₄O₂-SACs treatment.

Besides the pro-inflammatory factors, IL-17a, TNF- α , IL-12, and IL-23, how about the expression levels of typical anti-inflammatory factors, such as IL-10?

Though the authors found that psoriasis was ameliorated by upregulating ESR1, the pathogenesis of psoriasis is believed to be more closely related to STAT1 or NF- κ B (Nature 2007, 445, 866–873). Have the authors examined them accordingly?

What were the control group in relapse model? The IMQ-induced psoriasis mice treated by FeN₄O₂-SACs could be regarded as control group instead of healthy mice.

Some previous work could be cited and discussed if necessary.

Chemical Society Reviews, 2013, 42, 6060-6093.

Nature Communications, 2019, 10, 704.

Angewandte Chemie International Edition, 2022, 61, e202201101.

Reviewer #2:

Remarks to the Author:

In the manuscript by Xiangyu Lu et al., the authors have successfully devised a biomimetic approach to engineer an iron single-atom catalyst (FeN₄O₂-SACs). This remarkable feat has led to the development of a therapeutic modality with the potential to mitigate psoriasis and prevent relapses by restoring the estrogen receptor 1 (ESR1) through the catalyst's broad-spectrum reactive oxygen species (ROS) scavenging capability. The utilization of single-atom catalysts in this strategy is quite groundbreaking, and it represents a significant breakthrough in the field of anti-psoriasis research. However, before this article was published in Nature Communications, there were several important concerns that the authors needed to address.

Major comments

(1) In fact, the presence of the active center of Fe-N is not limited to antioxidant enzymes but extends to the catalytic sites of peroxidase-like enzymes (such as horseradish peroxidase, PDB: 6ATJ). How could the authors modulate the activity of the Fe-N catalytic site to selectively maintain antioxidant activity while attenuating peroxidase-like catalytic activities? Such knowledge may inform the design of novel therapeutic strategies aimed at ameliorating oxidative stress and mitigating the deleterious effects of reactive oxygen species.

(2) The administration of drugs in the treatment of psoriasis is a crucial aspect that demands careful attention. The method employed to administer the FeN₄O₂-SACs must be illustrated. If the topical application is utilized, whether the FeN₄O₂-SACs is capable of penetrating the skin barrier? Alternatively, if the oral or injectable route of administration is employed, it is essential to establish whether the FeN₄O₂-SACs can effectively reach the lesion status.

Minor comments

- (1) The content of Fe is very important for its catalytic activity, here's how the authors optimized the Fe content for its catalytic activity.
- (2) The authors chose natural enzymes and CeO₂, Mn₃O₄ nanoenzyme as a comparison to highlight the superior properties of catalytic activities of FeN₄O₂-SACs, why not other reported SACs antioxidant nanoenzyme for comparison.
- (3) The authors performed a systematic antioxidant catalytic activities in vitro study of FeN₄O₂-SACs, while the in vivo experiment was not involved relevant ROS evaluation, and the authors should explain.
- (4) The present study reports on the inhibitory effect of FeN₄O₂-SACs on HaCaT cell proliferation, as evidenced by an IC₅₀ value of 2.555 µg/ml. A comprehensive assessment of the cytotoxicity profile of FeN₄O₂-SACs on a variety of normal cell lines would help to clarify this issue and provide valuable insight into the potential clinical utility.
- (5) Fig.8c is confusing, is the relapse model of psoriasis originated from a psoriasis patients samples?
- (6) In my opinion, the immunohistochemical staining for the analysis of immune cells would be suboptimal and a more comprehensive and sophisticated flow cytometric method was suggested for a vivid visual representation and a quantitative measurement of immune cells.

Reviewer #3:

Remarks to the Author:

In this manuscript Xiangyu Lu and col address the activity of a new biomimetic catalyst (FeN₄O₂SACs) with antioxidant activity. In the first part this work, authors present the synthesis, structure, and characterization of FeN₄O₂SACs and during the second part, authors evaluate the role of FeN₄O₂SACs in HaCaT cell line, and in vivo using an animal model of psoriasis-like disease. ROS levels, proliferation and cytokine expression was evaluated in HaCaT cells stimulated with a mix of cytokines in the presence or not of FeN₄O₂SACs. On the other hand, administration FeN₄O₂SACs reduce epidermal hyperplasia, inflammatory infiltrate, and gene expression of pro-inflammatory cytokines. Based on the analysis of transcriptional profile from lesional skin before and after catalytic therapy, authors constructed a protein-protein interaction network and identified ESR1 as hub gene. To analyze the role of ESR1 in their model, authors performed silencing assays with shRNA in HaCaT cells. Finally, authors established what they call "relapse model of psoriasis" to evaluate the effect of FeN₄O₂SACs in the relapsing process.

One of the major concerns of this work is the effect of by FeN₄O₂SACs on cell viability. Indeed, dose of FeN₄O₂SACs was selected using a viability assay. Supplementary figure 9 show that the selected dose (2.5 µg/ml) induces 50% of cell death. Although FeN₄O₂SACs clearly have antioxidant activities as it is demonstrated in the first part of the manuscript, the role to inhibit the inflammatory response through ROS inhibition is not clear. In vitro experiments should be done using human primary keratinocytes and to demonstrate that the effect on ROS levels, proliferation and cytokine expression is not due to cell death.

In the in vivo model, authors show that FeN₄O₂SACs suppress the inflammation induced by a second dose of imiquimod after 12 days from the first application. In this sense, to clearly demonstrate the effect of FeN₄O₂SACs in a second challenge, its half- life should be analyzed. These experiments should be carried out using extended times (1 – 2 months) after the first stimuli. Tissue resident memory T cells play a key role in the psoriasis relapsing. What is the mechanism of FeN₄O₂SACs to avoid the inflammation after a second challenge of imiquimod? The role of ESR1 should be demonstrated in the in vivo model using conditional KO mice.

Figure 2-4, and supplementary figures 1-18 correspond to the synthesis, structure, and chemical characterization of FeN₄O₂SAC. Authors should summarize the information of these figures.

Figure 5, authors should clarify whether HaCaT cells were stimulated with M5 mix or IL-17 alone, at this point figures and text are confusing.

The mix of cytokines to mimic a psoriasis like environment should be composed only by IL-22, IL-17, IL-1a and TNFα, the rationale to include oncostatin M is not supported.

Figure 6. Group of mice treated alone with FeN4O2SACs should be included.

Regarding the protein-protein interaction network, a table with values of betweenness centrality, degree, clustering coefficient and log fold change from the main molecules should be included. Figure 7g is impossible to decipher Figure 7 should be restructured, some graphs could be omitted so that the protein network data can be shown more clearly.

Material and methods should be revised carefully to include the missing information like the confluency of HaCaT cells when were stimulated. Protocol for shESAR1 is also incomplete. Figure legends from supplementary figures are incomplete.

Figure 1 is missing

Response to reviewer I.

Comments from reviewer I:

Nanozymes are the functional nanomaterials with enzyme-like activities. They have attracted great attention recently, as evidenced that nanozymes were listed at Top Ten Emerging Techniques by IUPAS. Nanozymes have been explored for wide applications. Now, the authors developed a very interesting single atom catalyst with ROS scavenging for psoriasis therapy by upregulating ESR1. The catalytic therapy strategy was also proved effective in relapse prevention, which made nanomaterials with enzyme-like activities promising candidates for skin diseases. Therefore, I would suggest the publication after addressing the following concerns.

Response: Thank you very much for the positive comment and kind recommendation. Please find the following detailed responses to your suggestions.

1. Depart from CeO₂ and Mn₃O₄ depicted in Fig.4, other single atom catalyst should be provided to exhibit the advantages of FeN₄O₂-SACs if possible.

Response: Thank you for your valuable suggestions. According to your suggestion, we compared FeN₄O₂-SACs with commercially available cobalt-based, copper-based, and zinc-based single-atom catalysts, and the results show that FeN₄O₂-SACs have significantly higher antioxidant enzyme-mimic activities.

Following is our revisions to the manuscript:

Page 10 of Revised Manuscript (MS):

Finally, to evaluate the O₂^{•-} scavenging efficiency of FeN₄O₂-SACs, the commercial anti-inflammatory nanozymes (Mn₃O₄ and CeO₂) and metal single-atom catalysts (Co-SACs, Cu-SACs, Zn-SACs) were employed as controls⁴⁸⁻⁵¹. The results show that FeN₄O₂-SACs exhibit the highest SOD-like activity, and the activities of Fe in FeN₄O₂-SACs are 608, 334, and 369 times those of Ce in CeO₂, Mn in Mn₃O₄, and Zn in Zn-SACs, respectively (Fig. 4c, Supplementary Table 3).

Page 10-11 of Revised MS:

Fig. 4. CAT, SOD, and APX-like activities of FeN₄O₂-SACs. **a**, Schematic illustration of ROS clearance by FeN₄O₂-SACs. **b**, ESR spectra of FeN₄O₂-SACs and natural SOD for O₂⁻ clearance at pH=7.4 (X: xanthine; XO: xanthine oxidase). **c**, Comparison of SOD-like activities between FeN₄O₂-SACs and control materials CeO₂, Mn₃O₄, Co-SACs, Cu-SACs and Zn-SACs by WST-8 colorimetric analysis. **d**, CAT-like assay for eliminating varied concentrations of H₂O₂ using 125 μg/ml FeN₄O₂-SACs by UV absorption tests. **e**, CAT-like assay for eliminating 10 mM H₂O₂ using FeN₄O₂-SACs of varied concentrations; **f**, Lineweaver–Burk plots for FeN₄O₂-SACs and natural CAT at varied concentrations of H₂O₂ (0–20 mM). **g**, Comparison of the CAT-like activities between FeN₄O₂-SACs and control materials Mn₃O₄, CeO₂, Co-SACs, Cu-SACs and Zn-SACs. **h**, AsA characteristic absorption intensity decreases upon the additions of H₂O₂ of different samples. **i**, Comparison of the APX-like activities between FeN₄O₂-SACs and other materials Mn₃O₄ and CeO₂, Co-SACs, Cu-SACs and Zn-SACs.

Page 12 of Revised MS:

Finally, the catalytic activity of other commercially available single-atom catalysts in the decomposition of H₂O₂ was tested, further confirming the highest catalytic performance of FeN₄O₂-SACs in comparison with several reported nanozymes (Fig. 4g). Collectively, the above results demonstrate that FeN₄O₂-SACs possess excellent CAT-like enzymatic activity.

In addition, natural APX also catalyses the dissociation of H₂O₂ into H₂O using ascorbic acid (AsA) as the particular electron donor (Fig. 4a). Here, the APX-like activity was characterized by monitoring the absorbance of AsA at 290 nm. As Fig. 4h,i illustrates, FeN₄O₂-SACs exhibit a much higher activity in promoting the elimination of AsA than Co-SACs and Cu-SACs in the presence of H₂O₂, while Mn₃O₄, CeO₂ and Zn-SACs hardly exhibit such activities. The calculated AsA elimination activities of various materials are listed in Supplementary Table 3.

Page 45 of Revised Supplementary Information (SI):

Supplementary Table 3. Comparison of the enzymatic catalytic activities of FeN₄O₂-SACs with commercial antioxidant single-atom catalysts.

groups	SOD (units/mg)	CAT (units/mg)	APX (units/mg)
Fe in FeN ₄ O ₂ -SACs	21295.87	20828.59	354833.89
Ce in CeO ₂	34.99	9.82	27.73
Mn in Mn ₃ O ₄	63.61	44.22	16.85
Co in Co-SACs	2913.98	3830.22	77186.15
Cu in Cu-SACs	3792.52	2593.10	32619.05
Zn in Zn-SACs	57.67	88.12	240.15

2. In the method section, specifically in “Animal model”, it is recommended that additional details were provided, including the construction of the model and the technique employed for FeN₄O₂-SACs treatment.

Response: Thank you for your valuable suggestion. The details of the method utilized have been complemented with tracked changes.

Following is our revisions to the manuscript on Page 26-27 (“Animals and treatment” section) of Revised MS:

Male BALB/c mice aged 6-8 w were provided by Shanghai SLAC Laboratory Animal Co., Ltd., (no. 20220004020279, SYXK (Hu) 2018-0040). Imiquimod (IMQ) cream (H20030129, Sichuan Med-Shine Pharmaceutical Co.,Ltd) was used for psoriasis-like mouse model construction and all therapeutic strategies were performed for 8 days. All mice except those in the control group were topically administered 62.5 mg/mouse IMQ cream on the ears and backs once daily, and then different treatments involving PBS, FeN₄O₂-SACs or calcipotriol were administered to the same area 6 h later. For determination of FeN₄O₂-SACs consistency, mice were randomly divided into four intervention groups (62.5 mg/mouse IMQ cream and treatment 6 h later): Blank group (PBS, 80 μl/mouse), 1.25 μg/ml group (1.25 μg/ml of FeN₄O₂-SACs, 80 μl/mouse), 2.5 μg/ml group (2.5 μg/ml of FeN₄O₂-SACs, 80 μl/mouse), and 5 μg/ml group (5 μg/ml of FeN₄O₂-SACs, 80 μl/mouse). For therapeutic efficacy of FeN₄O₂-SACs, mice were randomly arranged into five groups: control group (PBS, 80 μl/mouse), FeN₄O₂-SACs group (2.5 μg/ml FeN₄O₂-SACs, 80 μl/mouse), IMQ group (PBS 6 h after IMQ), IMQ+FeN₄O₂-SACs group (2.5 μg/ml FeN₄O₂-SACs 6 h after IMQ), and IMQ+Cal group (1 mg/kg calcipotriol 6 h after IMQ). To evaluate the effectiveness of FeN₄O₂-SACs in decreasing recurrence, psoriasis relapse models were constructed. Mice were recovered for 30 days after 8-day treatment as mentioned above and then rechallenged with reduced doses of IMQ (20.8 mg/mouse) at the same lesion location for another 10 consecutive days. During the intervention period, the ear thickness of all mice was measured by vernier caliper, and the PASI score of lesion skins was calculated. Mice were executed by CO₂ inhalation suffocation at the indicated timepoint, and the lesion tissues were collected for further analysis (histology, RT-qPCR, flow cytometry, western blotting and RNA-seq). All the animal procedures were approved by the Ethics Committee of Shanghai University of Traditional Chinese Medicine (no. YYLAC-2022-160-3, Supporting Document 1).

3. Besides the pro-inflammatory factors, IL-17a, TNF-α, IL-12, and IL-23, how about

the expression levels of typical anti-inflammatory factors, such as IL-10?

Response: We appreciate your careful review. As suggested, the expression level of the anti-inflammatory factor *Il-10* was detected *in vivo*, and the results are shown below (Supplementary Fig 25).

Following is our revisions to the manuscript on Page 31 of Revised SI:

Supplementary Fig. 25. The mRNA expression of *Il-10* in the ear skin lesions of different groups, n=5 samples/group. ** $p < 0.01$ versus the control group; ### $p < 0.001$ versus the IMQ group.

4. Though the authors found that psoriasis was ameliorated by upregulating ESR1, the pathogenesis of psoriasis is believed to be more closely related to STAT1 or NF- κ B (Nature 2007, 445, 866–873). Have the authors examined them accordingly?

Response: Thank you very much for your comments. The levels of p-STAT1, p-STAT3 and NF- κ B p50 in lesional tissues were detected using immunohistochemical staining, and the results are shown in Fig. 6g,h. The reference (Nature 2007, 445, 866–873) has been cited as ref. [58].

Following is our revisions to the manuscript on Page 15-17 (“Alleviation of skin lesions in IMQ-induced psoriasis-like dermatitis mice” section) of Revised MS:

Immunohistochemistry staining (IHC) show that at the lesional skin, the increased numbers of CD3⁺ T cells, F4/80⁺ macrophages, and PCNA⁺ keratinocytes, as well as p-

STAT1⁺, p-STAT3⁺ and NF-κB p50⁺ signals⁵⁸ developed in the IMQ group have been strikingly diminished in the FeN₄O₂-SACs-treated and Cal groups, and the FeN₄O₂-SACs group exhibit a more significant reduction in CD3⁺ T cells, F4/80⁺ macrophages, p-STAT1⁺, p-STAT3⁺, NF-κB p50⁺ signals, CD3⁺ T cells and F4/80⁺ macrophages than the Cal group (Fig. 6g,h).

Fig. 6. g,h, Levels of inflammatory signals, T cells, macrophages, proliferative keratinocytes and tissue resident memory T cells analyzed by immunohistochemistry labelling of p-STAT1, p-STAT3, NF-κB p50, CD3, F4/80, PCNA and CD103 in the ear lesions on day 8 (g), and the corresponding quantifications (h). Scale bar=100 μm, n=4 samples/group. ****p*<0.001, ***p*<0.01, **p*<0.05 versus the IMQ group; ###*p*<0.001, ##*p*<0.01, #*p*<0.05 versus the IMQ+Cal group; ■■■*p*<0.001 versus the control group.

[58] Lowes, M. A., Bowcock, A. M. & Krueger, J. G. Pathogenesis and therapy of psoriasis. et al. Pathogenesis and therapy of psoriasis. Nature **445**, 866-73 (2007).

5. What were the control group in relapse model? The IMQ-induced psoriasis mice treated by FeN₄O₂-SACs could be regarded as control group instead of healthy mice.

Response: We truly appreciate your insightful comments. As suggested, the IMQ-induced psoriasis mice treated with FeN₄O₂-SACs but received no secondary IMQ induction have been included and regarded as the control group (Non-relapse+FeN₄O₂-

SACs). The results show that in 1-month (30 days) of recovery, the lesions of FeN₄O₂-SACs-treated mice (Non-relapse+FeN₄O₂-SACs) have recovered to the normal, while the Relapse+FeN₄O₂-SACs group shows no significant phenotypic changes even with secondary induction of IMQ. These results indicate that FeN₄O₂-SACs are capable of resisting the psoriasis relapse provoked by the secondary induction of IMQ.

Following is our revisions to the manuscript:

Page 21-23 of Revised MS:

Intriguingly, the skin of previously FeN₄O₂-SACs-treated mice exhibited mild psoriasiform lesions and ear thickness compared with the Non-relapse group (the 8-days IMQ-induced psoriasis mice received no secondary IMQ induction with or without FeN₄O₂-SACs treatment) (Fig. 8d-g and Supplementary Fig. 32). FeN₄O₂-SACs application has largely ameliorated IMQ-induced epidermal thickening and inflammatory infiltration on day 10 after the second stimulation compared with PBS/Cal. IHC results show that in contrast with the increased inflammatory cells and cell proliferation in the Relapse and Relapse+Cal groups, FeN₄O₂-SACs treatment significantly diminished the re-infiltration of p-STAT1⁺, p-STAT3⁺, NF-κB p50⁺ signals, CD3⁺ T cells, F4/80⁺ macrophages, and PCNA⁺ keratinocytes (Fig. 8h and Supplementary Fig. 33). With FeN₄O₂-SACs, the inflammatory cytokines *Il-17a*, *Tnf-α*, *Il-12*, *Il-23* were inhibited even with IMQ induction (Fig. 8i). The *in vivo* ROS detection showed that FeN₄O₂-SACs treatment significantly prevented the mice from the damage of oxidative stress (Supplementary Fig. 34). Additionally, the level of ESR1 was sustained by FeN₄O₂-SACs compared with the Relapse group (Supplementary Fig. 35). These data suggest that psoriasis catalytic therapy is an effective therapeutic modality for preventing disease rebounding of psoriasis-like skin lesions through ESR1, much better than conventional Cal treatment, in addition to its satisfactory efficacy in the initial treatments.

Fig. 8. **d**, Representative images of relapse models, $n=4$ samples/group. **e,f**, Quantifications of ear thickness (**e**) and PASI score (**f**), $n=4$ samples/group. **g**, Relative quantifications of histological staining. $n=4$ samples/group. **h**, Quantifications of $p\text{-STAT1}^+$, $p\text{-STAT3}^+$, $\text{NF-}\kappa\text{B } p50^+$ signals, CD3^+ T cells, F4/80^+ macrophages, and PCNA^+ keratinocytes in ear skin lesion tissues from different groups. $n=4$ samples/group. **i**, The mRNA expression levels of *Il-17a*, *Tnf- α* , *Il-12* and *Il-23* of ear skin tissues detected by RT-qPCR. $n=4$ samples/group. $***p<0.001$, $**p<0.01$, $*p<0.05$ versus Relapse group; $####p<0.001$, $###p<0.01$, $\#p<0.05$ versus Relapse+Cal group; $■■■p<0.001$ versus Non-relapse/Non-relapse+ $\text{FeN}_4\text{O}_2\text{-SACs}$ group.

Page 39-42 of Revised SI:

Supplementary Fig. 32. Histological staining of relapse models (ear and back). Scale

bar=100 μ m, n=4 samples/group.

Supplementary Fig. 33. IHC staining of p-STAT1, p-STAT3, NF-κB p50, CD3, F4/80 and PCNA in lesional ear skin tissues from different groups. Scale bar=100 μ m, n=4 samples/group.

Supplementary Fig. 34. a,b, The ROS level of relapsed or non-relapsed mouse ear lesions analyzed using fluorescence probe DHE (Dihydroethidium, Filter block of spORANGE at EX532-554/EM576-596 nm) (a) and the quantification (b) of relative mean fluorescence intensity (MFI), orange color indicates the ROS-positive, n=4 samples/group.

Supplementary Fig. 35. The protein expressions of ESR1 at the ear skin lesions of

different groups.

6. Some previous work could be cited and discussed if necessary.

Chemical Society Reviews, 2013, 42, 6060-6093.

Nature Communications, 2019, 10, 704.

Angewandte Chemie International Edition, 2022, 61, e202201101.

Response: Thank you for your kind suggestion. Related articles are representative work in the field of nanozymes and may provide new insights into the development of next-generation related nanomaterials. We have listed the corresponding references in the text.

[22] Wang, Q. et al. A Valence-Engineered Self-Cascading Antioxidant Nanozyme for the Therapy of Inflammatory Bowel Disease. *Angew. Chem. Int. Ed.* **61**, e202201101 (2022).

[24] Wei, H. & Wang, E. Nanomaterials with enzyme-like characteristics (nanozymes): next-generation artificial enzymes. *Chem. Soc. Rev.* **42**, 6060 (2013).

[25] Wang, X. et al. eg occupancy as an effective descriptor for the catalytic activity of perovskite oxide-based peroxidase mimics. *Nat. Commun.* **10**, 704 (2019).

Response to reviewer II.

Comments from reviewer II:

In the manuscript by Xiangyu Lu et al., the authors have successfully devised a biomimetic approach to engineer an iron single-atom catalyst (FeN₄O₂-SACs). This remarkable feat has led to the development of a therapeutic modality with the potential to mitigate psoriasis and prevent relapses by restoring the estrogen receptor 1 (ESR1) through the catalyst's broad-spectrum reactive oxygen species (ROS) scavenging capability. The utilization of single-atom catalysts in this strategy is quite groundbreaking, and it represents a significant breakthrough in the field of anti-psoriasis research. However, before this article was published in Nature Communications, there were several important concerns that the authors needed to address.

Response: Thank you very much for the insightful comments and suggestions. We have revised the manuscript carefully and provided detailed responses below in a point-by-point manner according to your suggestions.

Major comments

(1) In fact, the presence of the active center of Fe-N is not limited to antioxidant enzymes but extends to the catalytic sites of peroxidase-like enzymes (such as horseradish peroxidase, PDB: 6ATJ). How could the authors modulate the activity of the Fe-N catalytic site to selectively maintain antioxidant activity while attenuating peroxidase-like catalytic activities? Such knowledge may inform the design of novel therapeutic strategies aimed at ameliorating oxidative stress and mitigating the deleterious effects of reactive oxygen species.

Response: Thank you for the professional comment. The selectivity of oxidation/antioxidation is related to the valence state of the Fe sites based on previous researches. Generally, Fe^{2+} favors oxidation reactions (such as Fenton reactions), while Fe^{3+} accelerates reduction reactions. Previous work by Gu et al. suggests that the Fenton-like reaction triggered by Fe^{2+} in the Fe_3O_4 nanoenzymes is essential for its POD-like activity. After prolonged catalysis, Fe_3O_4 nanozymes suffer the phase transformation to $\gamma\text{-Fe}_2\text{O}_3$ with a depletable POD-like activity (Nature Communications, 2022, 13, 5365). In the work of An et al., the different antioxidant and physiological effects of three magnetite nanoparticles with different $\text{Fe}^{2+}/\text{Fe}^{3+}$ molar ratios were reported. These results demonstrate that Fe^{3+} enhances antioxidant activity and increases the anti-inflammatory effect of magnetite nanoparticles (ACS Appl. Bio Mater. 2021, 4, 1252–1267). In addition, Liu et al. reported that Fe^{3+} ions play a crucial role in antioxidant enzymes or nanozymes (ACS Cent. Sci. 2022, 8, 7–9). Yan et al further prepared iron single-atom nanozymes with abundant edge-hosted Fe-N₄ sites, where Fe has a chemical valence of +2 to +3, and possesses excellent catalase-like activity. Therefore, **increasing the proportion of Fe^{3+} in the Fe-N catalytic site is an effective way** to selectively maintain the antioxidant activity while attenuating the peroxidase-like catalytic activity. In our work, the Mössbauer data show that the

$\text{Fe}^{2+}/\text{Fe}^{3+}$ ratio in FeN_4O_2 -SACs is approximately 1:3, and the synchrotron radiation results show that the chemical valence of Fe is in between 2+ and 3+, which is conducive to selectively maintain antioxidant activity.

Following is our revisions to the manuscript on Page 7-8 of Revised Manuscript (MS):

Page 7:

The relative absorption area (74.4%) of D3 illustrates the significant presence of heme moieties in FeN_4O_2 -SACs. In addition, Fe in FeN_4O_2 -SACs is biased towards the 3+ valence state, which is conducive to selectively maintain antioxidant activity while attenuating peroxidase-like catalytic activities^{39,40}.

Page 8:

Normalized X-ray absorption near-edge structure (XANES) curves verify that the pre-edge position of FeN_4O_2 -SACs matches that of Fe_3O_4 , indicating that the Fe valency in FeN_4O_2 -SACs falls in between +2 and +3, similar to the results of the Mössbauer spectrum (Fig. 3b).

(2) The administration of drugs in the treatment of psoriasis is a crucial aspect that demands careful attention. The method employed to administer the FeN_4O_2 -SACs must be illustrated.

If the topical application is utilized, whether the FeN_4O_2 -SACs is capable of penetrating the skin barrier? Alternatively, if the oral or injectable route of administration is employed, it is essential to establish whether the FeN_4O_2 -SACs can effectively reach the lesion status.

Response: We appreciate your valuable comments and have supplemented the details of the method with the changes being highlighted. The FeN_4O_2 -SACs was topically applied onto the damaged skin of IMQ-induced mice at indicated concentration and 80 $\mu\text{l}/\text{mouse}$ (20 μl for each ear, 40 μl for back). The method employed on psoriasis treatments can be refer to *Biomaterials* 2021, 276, 121027^[1] and *Sci Adv* 2020, 6, eabb5274^[2]. In this study of FeN_4O_2 -SACs application in repairing the skin lesions, the

penetration into the skin barrier was not considered. As the oral or injection administrations were not employed here, therefore the side effects associated with these routes of administrations were not taken into account. Additionally, the organ toxicity of FeN₄O₂-SACs in the topical application to psoriasis-like dermatitis was evaluated, and no safety concerns can be seen (Supplementary Fig. 26).

- [1] Yan, Y., Liang, H., Liu, X., Liu, L. & Chen, Y. Topical cationic hairy particles targeting cell free DNA in dermis enhance treatment of psoriasis. *Biomaterials* **276**, 121027 (2021).
- [2] Liang, H. et al. Topical nanoparticles interfering with the DNA-LL37 complex to alleviate psoriatic inflammation in mice and monkeys. *Sci Adv* **6**, eabb5274 (2020).

Following is our revisions to the manuscript on Page 32-33 of Revised Supplementary Information (SI):

Supplementary Fig. 26. The organ toxicity detections of FeN₄O₂-SACs. **a**, The hepatic function (alanine aminotransferase, aspartate aminotransferase, total bilirubin,

direct bilirubin, and indirect bilirubin), renal function (urea nitrogen, creatinine, serum β 2-microglobulin, and serum uric acid), and glycolipid metabolism indexes (total cholesterol, triglyceride, high density lipoprotein cholesterol, low density lipoprotein cholesterol, fasting blood glucose, and insulin) detected by ELISA, n=4 samples/group. **b**, The histopathology of liver, spleen and kidney before and after treatments. n=3 samples/group. *** p <0.001, ** p <0.001, * p <0.001; #### p <0.001, ## p <0.01, # p <0.05; ns means no significance.

Following is our revisions to the manuscript on Page 26-27 (“Animals and treatment” section) of Revised MS.

Male BALB/c mice aged 6-8 w were provided by Shanghai SLAC Laboratory Animal Co., Ltd., (no. 20220004020279, SYXK (Hu) 2018-0040). Imiquimod (IMQ) cream (H20030129, Sichuan Med-Shine Pharmaceutical Co.,Ltd) was used for psoriasis-like mouse model construction and all therapeutic strategies were performed for 8 days. All mice except those in the control group were topically administered 62.5 mg/mouse IMQ cream on the ears and backs once daily, and then different treatments involving PBS, FeN₄O₂-SACs or calcipotriol were administered to the same area 6 h later. For determination of FeN₄O₂-SACs consistency, mice were randomly divided into four intervention groups (62.5 mg/mouse IMQ cream and treatment 6 h later): Blank group (PBS, 80 μ l/mouse), 1.25 μ g/ml group (1.25 μ g/ml of FeN₄O₂-SACs, 80 μ l/mouse), 2.5 μ g/ml group (2.5 μ g/ml of FeN₄O₂-SACs, 80 μ l/mouse), and 5 μ g/ml group (5 μ g/ml of FeN₄O₂-SACs, 80 μ l/mouse). For therapeutic efficacy of FeN₄O₂-SACs, mice were randomly arranged into five groups: control group (PBS, 80 μ l/mouse), FeN₄O₂-SACs group (2.5 μ g/ml FeN₄O₂-SACs, 80 μ l/mouse), IMQ group (PBS 6 h after IMQ), IMQ+FeN₄O₂-SACs group (2.5 μ g/ml FeN₄O₂-SACs 6 h after IMQ), and IMQ+Cal group (1 mg/kg calcipotriol 6 h after IMQ). To evaluate the effectiveness of FeN₄O₂-SACs in decreasing recurrence, psoriasis relapse models were constructed. Mice were recovered for 30 days after 8-day treatment as mentioned above and then rechallenged with reduced doses of IMQ (20.8 mg/mouse) at the same lesion location for another 10 consecutive days. During the intervention period, the ear thickness of all mice was

measured by a vernier caliper, and the PASI score of lesion skins was calculated. Mice were executed by CO₂ inhalation suffocation at the indicated timepoint, and the lesion tissues were collected for further analysis (histology, RT-qPCR, flow cytometry, western blotting and RNA-seq). All the animal procedures were approved by the Ethics Committee of Shanghai University of Traditional Chinese Medicine (no. YYLAC-2022-160-3, Supporting Document 1).

Minor comments

(1) The content of Fe is very important for its catalytic activity, here's how the authors optimized the Fe content for its catalytic activity.

Response: We appreciate your insightful comments. In this work, we ended up using FeN₄O₂-SACs where the original molar percentage of Fe adding was 1% of the total molar amount of Zn and Fe. Initially, we also prepared catalytic materials with Fe adding amounts of 2% and 5%. However, XRD patterns (**Fig. 1**) preliminarily indicate that the 5% Fe-added material is partially crystalline and in the form of iron clusters, while the 1% and 2% added material may remain fully amorphous. Then, X-ray absorption fine structure (XAFS) spectroscopy was employed to further characterize the 1% added catalysts, the existence of Fe-Fe bonds means the presence of Fe clusters addition to the single Fe atoms. These results indicate that metal clusters are easily generated during the preparation of single-atom catalysts and that the presence of clusters may reduce the atomic utilization^[1,2]. If the amount of Fe doping is further reduced, the number of active sites may decrease to a very low percentage, which will bring about difficulties in characterization. Therefore, we chose 1% Fe-added material as our target and removed the metal clusters by sulfuric acid pickling to prepare FeN₄O₂-SACs with 100% atomic dispersion. The Fe concentration in single Fe atom form in FeN₄O₂-SACs is 0.43%, which is not a high doping amount. Therefore, we will explore a new synthetic method to optimize the Fe doping amount and prepare single-atom catalysts with high iron loading (>5%) in our next work. Thank you very much again for your insightful suggestions.

Fig. 1. XRD patterns of NC and Fe single-atom catalysts with varying Fe doping.

- [1] Zhao, L., Zhang, Y., Huang, LB. et al. Cascade anchoring strategy for general mass production of high-loading single-atomic metal-nitrogen catalysts. *Nat Commun* **10**, 1278 (2019).
- [2] Jiao, L., Zhang, R., Wan, G. et al. Nanocasting SiO₂ into metal–organic frameworks imparts dual protection to high-loading Fe single-atom electrocatalysts. *Nat Commun* **11**, 2831 (2020).

(2) The authors chose natural enzymes and CeO₂, Mn₃O₄ nanoenzyme as a comparison to highlight the superior properties of catalytic activities of FeN₄O₂-SACs, why not other reported SACs antioxidant nanoenzyme for comparison.

Response: Thank you very much for pointing out this issue. According to your suggestion, we compared the catalytic activities of FeN₄O₂-SACs with other SACs antioxidant nanoenzymes reported in the literature, and the relevant results are listed in Supplementary Table 3, demonstrating the superiority of the catalytic activity of FeN₄O₂-SACs.

Following is our revisions to the manuscript:

Page 10 of Revised MS:

In addition, compared with other antioxidant nanoenzymes of SACs reported in the literatures (Supplementary Table 4), such as single-atom Ir enzyme mimics (Ir NC SAzymes)⁵⁸ and single-atom cobalt nanozymes (Co-SAzymes)⁵⁹, FeN₄O₂-SACs exhibit superior catalytic activities.

Page 46 of Revised SI:

Supplementary Table 4. Comparison of the enzymatic catalytic activities of FeN₄O₂-SACs with other reported SACs antioxidant nanoenzyme.

Materials	ICP	SOD (units/mg)	CAT (units/mg)	APX (units/mg)
Fe in FeN ₄ O ₂ -SACs	0.43%	21295.87	20828.59	354833.89
Ir in Ir NC SAzymes ⁶	0.6%	-	9728.33	-
Fe in Fe-SANzyme ⁷	0.45%	-	11697.78	-
Rh in RhN ₄ nanozymes ⁸	0.19%	574.29	50306.09	-
V in VN ₄ nanozymes ⁸	0.22%	803.92	12176.47	-

(3) The authors performed a systematic antioxidant catalytic activities in vitro study of FeN₄O₂-SACs, while the in vivo experiment was not involved relevant ROS evaluation, and the authors should explain.

Response: Thank you very much for your professional recommendation. Supplementary animal experiment was repeated and the *in vivo* ROS levels were evaluated. Consistent with the *in vitro* study, FeN₄O₂-SACs treatment has been found to significantly ameliorate the oxidative stress induced by IMQ (Fig. 6j,k and Supplementary Fig. 34).

Following is our revisions to the manuscript:

Page 26 of Revised MS:

***In vivo* ROS detection**

The *in vivo* ROS level was analyzed using BBoxiProbe® DHE ROS detection kit (BB-47051). According to the manufacturer's instructions, the fluorescence to determine the

ROS level in cells was detected by fluorescence microscope at the excitation wavelength of 532-544 nm and the emission wavelength of 576-596 nm. The filter block we used was spOrange, and the intensity of orange fluorescence is proportional to the level of ROS.

Page 16-17 of Revised MS:

Fig. 6. j,k, The ROS level of ear skins from different groups detected by fluorescence probe DHE (Dihydroethidium, Filter block of spORANGE at EX532-554/EM576-596 nm) (j) and the quantification (k) of relative mean fluorescence intensity (MFI), orange color indicates the ROS-positive, n=4 samples/group.

Page 41 of Revised SI:

Supplementary Fig. 34. a,b, The ROS level of relapsed or non-relapsed mouse ear lesions analyzed using fluorescence probe DHE (Dihydroethidium, Filter block of spORANGE at EX532-554/EM576-596 nm) (a) and the quantification (b) of relative mean fluorescence intensity (MFI), orange color indicates the ROS-positive, n=4 samples/group.

(4) The present study reports on the inhibitory effect of FeN₄O₂-SACs on HaCaT cell proliferation, as evidenced by an IC₅₀ value of 2.555 µg/ml. A comprehensive assessment of the cytotoxicity profile of FeN₄O₂-SACs on a variety of normal cell lines would help to clarify this issue and provide valuable insight into the potential clinical utility.

Response: We truly appreciate your valuable comments. As suggested, the cytotoxicity profile of FeN₄O₂-SACs on normal human epidermal keratinocytes (NHEK cells) was

determined, and the inhibitory effect of FeN₄O₂-SACs on NHEK cell proliferation was also confirmed. The concentration of 2.5 μg/ml showed significant inhibitions on cell proliferation and inflammation in NHEK cells.

Following is our revisions to the manuscript:

Page 13 of Revised MS:

Additionally, the cytotoxicity profile of FeN₄O₂-SACs on normal human epidermal keratinocytes (NHEK cells) was also evaluated, and anti-proliferation and anti-inflammation effects can be seen (Supplementary Fig. 20).

Page 26 of Revised SI:

Supplementary Fig. 20. FeN₄O₂-SACs effectively inhibit the hyperproliferation and inflammation of NHEK cells. **a**, The cytotoxicity profile of FeN₄O₂-SACs on NHEK cells (48 h). Data from 3 separate experiments as mean ± SD. **b**, CCK-8 assay showing the cell proliferative viabilities of different groups. Data from 3 separate experiments as mean ± SD. **c**, RT-qPCR results for the inflammatory factors *TNF-α*, *IL-6*, and *IL-8*. Data from 5 separate experiments as mean ± SD. ****p*<0.001, **p*<0.05 versus the Control group; ###*p*<0.001, #*p*<0.05 versus the M5 group.

(5) Fig.8c is confusing, is the relapse model of psoriasis originated from a psoriasis patients samples?

Response: Thank you very much for pointing out this issue. The relapse model of psoriasis was established on mouse samples, not patient samples. The psoriasis patient in Fig. 8c is to display the clinical relevance of the target ESR1 to relapse (demonstrated by the IHC results in Fig. 8a,b), which shows that the ESR1 level is downregulated in

the lesion of relapse patient. Therefore, we further verified whether FeN₄O₂-SACs prevented psoriasis would relapse or not in mouse samples. Following your suggestion, we have modified **Fig.8c** as shown below to avoid confusion.

Following is our revisions to the manuscript on Page 22 (Fig. 8) of Revised MS.

Fig 8. c, Experimental design of IMQ-stimulated psoriasis relapse model.

(6) In my opinion, the immunohistochemical staining for the analysis of immune cells would be suboptimal and a more comprehensive and sophisticated flow cytometric method was suggested for a vivid visual representation and a quantitative measurement of immune cells.

Response: Thank you for your comments. We utilized a flow cytometric method to analyze the CD3⁺ T lymphocytes, F4/80⁺ macrophages and CD103⁺ tissue resident memory T cells at skin lesions from different groups, and the results were consistent with the immunohistochemical staining.

Following is our revisions to the manuscript:

Page 5 of Revised SI:

Flow cytometry

The isolation of cells from the skin tissues of mice referred to the previous method². For flow cytometry, cells were stained with fluorescently labeled antibodies directly after cell processing. For experiments with murine tissues, the following mAbs were used for extracellular staining: CD45-BV421(30-F11, Biolegend, 103134), CD3-FITC (17A2, Biolegend, 100204), CD4-PerCP (RM405, Biolegend, 100538), CD8-PE(53-6.7, Biolegend, 100708), F4/80-APC (BM8, Invitrogen, 17-4801-82), CD11b-BV785

(M1/70, Biolegend, 101243), Viability-APC-Cy7 (Invitrogen, 65-0865-14), CD69-APC (H1.2F3, Biolegend, 104512), CD103-BV510 (2E7, Biolegend, 121423). Samples were acquired on the BD LSR Fortessa X-20 equipped with four lasers (BD Biosciences) and data were analyzed using FlowJo software (Ashland, OR).

Page 15 of Revised MS:

Consistently, the flow cytometry results showed that FeN₄O₂-SACs remarkably reduced the CD3⁺T cells, F4/80⁺CD11b⁺macrophages and CD103⁺CD69⁺T_{RM} cells (Supplementary Fig. 24).

Page 30 of Revised SI:

Supplementary Fig. 24. FeN₄O₂-SACs remarkably reduce the numbers of CD3⁺ T lymphocytes, CD11b⁺F4/80⁺ macrophages and CD103⁺CD69⁺T_{RM} cells at the ear skins. **a**, The percentage of CD3⁺ T lymphocytes in different groups determined by flow cytometry. n=6 samples/group. **b**, The percentage of CD11b⁺F4/80⁺ macrophages (in CD45⁺ cells) in different groups quantified by flow cytometry. n=6 samples/group. **c**, The percentage of CD103⁺CD69⁺T_{RM} cells (in CD3⁺ T lymphocytes) in different groups quantified by flow cytometry. n=6 samples/group. ****p*<0.001, ***p*<0.01 versus

the IMQ group; ^{###} $p < 0.001$ versus the IMQ+Cal group; ^{■■■} $p < 0.001$ versus the control group.

Response to reviewer III.

Comments from reviewer III:

In this manuscript Xiangyu Lu and col address the activity of a new biomimetic catalyst (FeN₄O₂SACs) with antioxidant activity. In the first part this work, authors present the synthesis, structure, and characterization of FeN₄O₂SACs and during the second part, authors evaluate the role of FeN₄O₂SACs in HaCaT cell line, and in vivo using an animal model of psoriasis-like disease. ROS levels, proliferation and cytokine expression was evaluated in HaCaT cells stimulated with a mix of cytokines in the presence or not of FeN₄O₂SACs. On the other hand, administration FeN₄O₂SACs reduce epidermal hyperplasia, inflammatory infiltrate, and gene expression of pro-inflammatory cytokines. Based on the analysis of transcriptional profile from lesional skin before and after catalytic therapy, authors constructed a protein-protein interaction network and identified ESR1 as hub gene. To analyze the role of ESR1 in their model, authors performed silencing assays with shRNA in HaCaT cells. Finally, authors established what they call “relapse model of psoriasis” to evaluate the effect of FeN₄O₂SACs in the relapsing process.

Response: Thank you very much for the professional comments and summary of the article. Please find the following detailed responses to your suggestions.

1. One of the major concerns of this work is the effect of by FeN₄O₂SACs on cell viability. Indeed, dose of FeN₄O₂SACs was selected using a viability assay. Supplementary figure 9 show that the selected dose (2.5 ug/ml) induces 50% of cell death. Although FeN₄O₂SACs clearly have antioxidant activities as it is demonstrated in the first part of the manuscript, the role to inhibit the inflammatory response through ROS inhibition is not clear. In vitro experiments should be done using human primary keratinocytes and to demonstrate that the effect on ROS levels, proliferation and cytokine expression is not due to cell death.

Response: Thank you for the professional comment. We used normal human epidermal keratinocytes (NHEK cells) and found that FeN₄O₂-SACs could effectively reduce the cell proliferation and inflammation, as well as scavenge ROS *in vitro*.

Following is our revisions to the manuscript:

Page 13 of Revised Manuscript (MS):

Additionally, the cytotoxicity profile of FeN₄O₂-SACs on normal human epidermal keratinocytes (NHEK cells) was also evaluated, and anti-proliferation and anti-inflammation effects can be seen (Supplementary Fig. 20) (Page 13, lines 4-6).

Page 26 of Revised Supplementary Information (SI):

Supplementary Fig. 20. FeN₄O₂-SACs effectively inhibit the hyperproliferation and inflammation of NHEK cells. **a**, The cytotoxicity profile of FeN₄O₂-SACs on NHEK cells (48 h). Data from 3 separate experiments as mean ± SD. **b**, CCK-8 assay showing the cell proliferative viabilities of different groups. Data from 3 separate experiments as mean ± SD. **c**, RT-qPCR results for the inflammatory factors *TNF-α*, *IL-6*, and *IL-8*. Data from 5 separate experiments as mean ± SD. ****p*<0.001, **p*<0.05 versus the Control group; ###*p*<0.001, #*p*<0.05 versus the M5 group.

Page 13 of Revised MS:

Similarly, reductions in ROS levels were observed in NHEKs by flow cytometry (Supplementary Fig. 22), which is thought to be advantageous to psoriasis therapy (Page 13, line 14-15).

Page 28 of Revised SI:

Supplementary Fig. 22. a,b, Flow cytometry analysis of NHEKs cells in different groups stained by DCFH-DA (2,7-Dichlorodihydrofluorescein diacetate, EX498/EM530 nm) (a) and the quantification (b) of relative mean fluorescence intensity (MFI), n=3 samples/group. Data from 3 separate experiments as mean ± SD.

2. In the *in vivo* model, authors show that FeN4O2SACs suppress the inflammation induced by a second dose of imiquimod after 12 days from the first application. In this sense, to clearly demonstrate the effect of FeN4O2SACs in a second challenge, its half-life should be analyzed. These experiments should be carried out using extended times (1–2 months) after the first stimuli. Tissue resident memory T cells play a key role in the psoriasis relapsing. What is the mechanism of FeN4O2SACs to avoid the inflammation after a second challenge of imiquimod? The role of ESR1 should be demonstrated in the *in vivo* model using conditional KO mice.

Response: We appreciate your insightful comments. All the *in vivo* experiments have been reconducted, and the results have been updated. In the relapse experiment, the second dose of imiquimod was applied 1 month after the first stimuli as you suggested. Additionally, a marker of tissue resident memory T cells (CD103) was detected by immunohistochemical staining and flow cytometry. The results reveal that FeN4O2-SACs effectively reduce the activation of tissue resident memory T cells. To demonstrate the role of ESR1, the conditional KO mice were preferred. However, considering that the experimental periodicity would be longer than 1.5 years, we would like to perform this experiment in the future. Additionally, since psoriasis is mainly

characterized by inflammation and hyperproliferation of keratinocytes, we applied CRISPR/Cas 9 technique to knock out the expression of ESR1 in HaCaT cells (ESR1-KO) to investigate the role of ESR1 in FeN₄O₂-SACs treatments. In ESR1-KO HaCaT cells, the curative effect of FeN₄O₂-SACs in ameliorating inflammation and hyperproliferation was blocked, indicating that FeN₄O₂-SACs function through ESR1.

Following is our revisions to the manuscript:

Page 20 of Revised MS:

Conversely, when the expression of ESR1 was knocked down using CRISPR/Cas9, FeN₄O₂-SACs treatment was not able to rescue the hyperproliferation and undue inflammation of M5-induced psoriasis-like *in vitro* model, indicating that ESR1 is indispensable for the curative effects of FeN₄O₂-SACs (Supplementary Fig. 29).

Page 36 of Revised SI:

Supplementary Fig. 29. FeN₄O₂-SACs inhibit the proliferation and inflammation of HaCaT cells through ESR1. **a,b**, Knockout (KO) validation of ESR1 in HaCaT cells by sequencing (**a**) and western blotting (**b**). **c**, CCK-8 assay showing the cell proliferative viabilities regulated by ESR1-KO and the effect of FeN₄O₂-SACs. Data from 3 separate experiments as mean ± SD. **d**, RT-qPCR results for the inflammatory

factors *TNF- α* , *IL-6*, and *IL-8*. Data from 5 separate experiments as mean \pm SD. *** $p < 0.001$ versus the Control group; ### $p < 0.001$, ## $p < 0.01$ versus the M5 group; ■■■ $p < 0.001$, ■ $p < 0.05$ versus the M5+ FeN₄O₂-SACs group.

3. Figure 2-4, and supplementary figures 1-18 correspond to the synthesis, structure, and chemical characterization of FeN₄O₂SAC. Authors should summarize the information of these figures.

Response: Thank you very much for your kind comment. According to your suggestion, the information of these figures has been summarized and supplemented in Page 12.

Following is our revisions to the manuscript on Page 12 of Revised MS:

In summary, we have demonstrated that FeN₄O₂-SACs manifest a homogeneous, amorphous, dodecahedral structure with single Fe-N₄O₂ active sites being embedded in the structure, which endows the materials with remarkable or significant SOD-, CAT-, and APX-like activities. Such activities are greatly higher than those of typical anti-inflammatory nanozymes (Mn₃O₄ and CeO₂) and commercial Co-SACs, Cu-SACs, and Zn-SACs, even better than natural enzymes under some reaction conditions. In addition, compared with other antioxidant nanozymes of SACs reported in the literature (Supplementary Table 4), such as single-atom Ir enzyme mimics (Ir NC SAzymes)⁵⁸ and single-atom cobalt nanozymes (Co-SAzymes)⁵⁹, FeN₄O₂-SACs exhibit superior catalytic activities. It can be inferred that the multienzyme activity is related to the fact that FeN₄O₂-SACs have a structure analogous to those of natural anti-inflammatory enzymes.

4. Figure 5, authors should clarify whether HaCaT cells were stimulated with M5 mix or IL-17 alone, at this point figures and text are confusing.

The mix of cytokines to mimic a psoriasis like environment should be composed only by IL-22, IL-17, IL-1 α and TNF α , the rationale to include oncostatin M is not supported.

Response: We appreciate your valuable comments. In formal experiments, all the HaCaT cells were stimulated with M5 mix instead of IL-17 alone and the corresponding

correction has been made. In the mix of cytokines to mimic a psoriasis-like environment, the composition of IL-22, IL-17, IL-1 α , TNF- α and oncostatin M can be found in references^{64,65}. Meanwhile, CCK-8 and RT-qPCR assays were conducted to determine whether oncostatin M should be included, and the mixture with oncostatin M shows a higher cell proliferation rate and elevated inflammatory factors than that without oncostatin M.

a, Proliferation of HaCaT cells with or without oncostatin M. Data from 3 separate experiments as mean \pm SD. **b**, Expressions of inflammatory factors *TNF- α* , *IL-6* and *IL-8* with or without oncostatin M. Data from 4 separate experiments as mean \pm SD.

[64] Kuai, L. et al. Celastrol Attenuates Psoriasiform Inflammation by Targeting the IRF1/GSTM3 Axis. *J. Invest. Dermatol.* **142**, 2281-2285. e11 (2022).

[65] Li, C. J. et al. Cornulin Is Induced in Psoriasis Lesions and Promotes Keratinocyte Proliferation via Phosphoinositide 3-Kinase/Akt Pathways. *J. Invest. Dermatol.* **139**, 71-80 (2019).

5. Figure 6. Group of mice treated alone with FeN4O2SACs should be included.

Response: Thank you very much for your valuable suggestion. A group of mice treated alone with FeN4O₂-SACs has been included as suggested (Fig. 5).

Following is our revisions to the manuscript on Page 16-17 of Revised MS:

Fig. 6. FeN₄O₂-SACs effectively alleviate the psoriasis dermatitis *in vivo*. **a**, Schemes of the IMQ-induced psoriasiform model, treatments and efficacy. **b**, Representative images of lesions on day 8, n=4 samples/group. **c,d**, Quantification of ear thickness (**c**) and PASI score (**d**). **e,f**, Histological staining (**e**) and the corresponding quantifications (**f**). Scale bar=100 µm, n=4 samples/group. **g,h**, Levels of inflammatory signals, T cells, macrophages, proliferative keratinocytes and tissue resident memory T

cells analyzed by immunohistochemistry labelling of p-STAT1, p-STAT3, NF-κB p50, CD3, F4/80, PCNA and CD103 in the ear lesions on day 8 (g), and the corresponding quantifications (h). Scale bar=100 μm, n=4 samples/group. i, The mRNA expression levels of *Il-17a*, *Tnf-α*, *Il-12*, and *Il-23* at the ear skin tissues detected by RT-qPCR. n=4 samples/group. j,k, The ROS level of ear skins from different groups detected by fluorescence probe DHE (Dihydroethidium, Filter block of spORANGE at EX532-554/EM576-596 nm) (j) and the quantification (k) of relative mean fluorescence intensity (MFI), orange color indicates positive, n=4 samples/group. *** $p<0.001$, ** $p<0.01$, * $p<0.05$ versus the IMQ group; ### $p<0.001$, ## $p<0.01$, # $p<0.05$ versus the IMQ+Cal group; ■■■ $p<0.001$ versus the control group.

6. Regarding the protein-protein interaction network, a table with values of betweenness centrality, degree, clustering coefficient and log fold change from the main molecules should be included. Figure 7g is impossible to decipher Figure 7 should be restructured, some graphs could be omitted so that the protein network data can be shown more clearly.

Response: We sincerely appreciate your professional suggestion. The protein-protein interaction network, as depicted in Figure 7g, has been restructured.

Following is our revisions to the manuscript on Page 18 of Revised MS:

Fig. 7. g, Protein network of the DE mRNAs.

7. Material and methods should be revised carefully to include the missing information like the confluency of HaCaT cells when were stimulated. Protocol for shESAR1 is also incomplete. Figure legends from supplementary figures are incomplete.

Response: Thanks a lot for the suggestion. As suggested, we have revised the materials and methods carefully with the changes being highlighted.

Following is our revisions to the materials and methods on Page 3-4 of Revised SI:

To knock down the expression of ESR1 (RefSeq NM_000125) in HaCaT cells, cells were cultured in puromycin (1 µg/ml) and co-transfected with shRNA-control vectors or shRNA-ESR1 vectors in accordance with the manufacturer's guidelines. The shRNA-ESR1 sequences were obtained from Sigma (<https://www.sigmaaldrich.cn/CN/zh/semi->

configurators/shrna?activeLink=selectClones): pLKO.1puro-shhESR1-1F,
CCGGGCCCTACTACCTGGAGAACGACTCGAGTCGTTCTCCAGGTAGTAGGG
CTTTTTG; pLKO.1puro-shhESR1-1R,

AATTCAAAAAGCCCTACTACCTGGAGAACGACTCGAGTCGTTCTCCAGGTA
GTAGGGC; pLKO.1puro-shhESR1-2F,

CCGGCTACAGGCCAAATTCAGATAACTCGAGTTATCTGAATTTGGCCTGTAG
TTTTTG; pLKO.1puro-shhESR1-2R,

AATTCAAAAAGCCCTACTACCTGGAGAACGACTCGAGTTATCTGAATTTGG
CCTGTAG; pLKO.1puro-shhESR1-3F,

CCGGAGCACCTGAAGTCTCTGGAATCGAGTTCCAGAGACTTCAGGGTGC
TTTTTTG; pLKO.1puro-shhESR1-3R,

AATTCAAAAAGCACCTGAAGTCTCTGGAATCGAGTTCCAGAGACTTC
AGGGTGCT. The sequences were cloned into the lentivirus vector pLKO.1 puro with

shRNA construct. The enzyme digestion site for directed cloning was 5'-AgeI/EcoRI-3'. The knock-down efficacy of shhESR1-1 was the highest, therefore shhESR1-1 was chosen for the following experiments.

8. Figure 1 is missing

Response: Thank you very much for pointing out this issue. We have modified the picture title from "Scheme 1" to "Fig. 1".

Reviewers' Comments:

Reviewer #1:

Remarks to the Author:

The authors have addressed my concerns.

Reviewer #2:

Remarks to the Author:

The authors' conscientious efforts in addressing the reviewer's concerns, as well as their successful expansion and enrichment of the original passage, warrant the publication of this manuscript.

Reviewer #3:

Remarks to the Author:

After review the revised version, some concerns remains and should be addressed.

One of the major concern during the first revision was the cell death induced by the compound (50%, supplementary figure 19). This is very important for the in vitro assays. Indeed, CCK8 assay evaluate cell viability, authors use this assay to analyze proliferation. BrdU incorporation assay is more advisable.

An additional important concern is effect of FeN4O2-SACs in psoriasis relapse. Data from TRM cells are not convincing. Authors show 70-80% of CD69+ CD103+ from the CD3+ population under control condition which is not in line with images of Figure 6g. Murine resident memory T cells are mainly at epidermis. Identification of TRM cells should be done using immunofluorescence using anti-CD8 and anti-CD103. *J Invest Dermatol.* 2020 Apr; 140(4): 748–755.

How long does the antioxidant effect of the compound last once it has been applied to the skin? Supplementary figure 20. Panel a, Y axis indicate proliferative level but cell viability, authors quote at figure legend and results that this figure correspond to cytotoxicity level, please clarify this point.

Supplementary figure 22. Gating strategy should be included to demonstrate that death cell are excluded from the analysis. Also, M5 instead IL-17 should indicated.

Figure 5. Please include gating strategy, it is important to confirm that cell viability dye has been used to exclude death cells from the analysis

Supplementary figure 24. It is mandatory to include a cell viability dye for these assays.

Reviewer #1 (Remarks to the Author):

The authors have addressed my concerns.

Response: Thank you very much for the positive comment.

Reviewer #2 (Remarks to the Author):

The authors' conscientious efforts in addressing the reviewer's concerns, as well as their successful expansion and enrichment of the original passage, warrant the publication of this manuscript.

Response: Thank you very much for the positive comment and kind recommendation.

Reviewer #3 (Remarks to the Author):

After review the revised version, some concerns remains and should be addressed.

Response: Thank you very much for the kind suggestion. Please find the following detailed responses to your suggestions.

1. One of the major concern during the first revision was the cell death induced by the compound (50%, supplementary figure 19). This is very important for the in vitro assays. Indeed, CCK8 assay evaluate cell viability, authors use this assay to analyze proliferation. BrdU incorporation assay is more advisable.

Response: We truly appreciate your suggestions. Indeed CCK-8 generally reflects the activity of cells, it can also be used to detect cell proliferation as references [1,2]. Additionally, the thymine nucleoside analogs, such as BrdU and EdU, are usually applied for cell proliferation detection. EdU antibody is 1/500 the size of the BrdU antibody, making it easier to spread within cells and does not require a strict sample denaturation, which can be more sensitive and convenient. In this revision, EdU incorporation assay has been conducted, and the results are shown as follows.

[1] Du, A. S. et al. M6A-mediated upregulation of circMDK promotes tumorigenesis and acts as a nanotherapeutic target in hepatocellular carcinoma. *Mol. Cancer* **21**, 109 (2022).

[2] Liu, X. et al. CircMYH9 drives colorectal cancer growth by regulating serine metabolism and redox homeostasis in a p53-dependent manner. *Mol. Cancer* **20**, 114 (2021).

The following is our revisions to the manuscript:

On Page 13 of the Revised Manuscript, which reads:

Additionally, the cytotoxicity profile of FeN₄O₂-SACs on normal human epidermal keratinocytes (NHEK cells) was also evaluated, and its anti-proliferation and anti-inflammation effects can be clearly seen (Supplementary Fig. 20-21).

On page 4 of Revised Supplementary Information, which reads:

EdU assay

BeyoClick™ EdU cell proliferation detection kit (TMB method) was used, and experiment was performed according to the manufacturer's instruction. Briefly, cells in a 96 well plate were cultured overnight to normal state and then, different treatments were applied (M5 or M5+FeN₄O₂-SACs) for 72 hours. TMB color solution (0.1 ml) was added and incubated at room temperature for 5-30 minutes. The absorbance was measured at 370 nm.

And on page 26 of Revised Supplementary Information as Supplementary Figure 20, which is:

Supplementary Fig. 20. FeN₄O₂-SACs effectively inhibit the cell proliferation of HaCaT cells (a) and NHEK cells (b) by EdU assay. Data from 4 separate experiments as mean ± SD.

2. An additional important concern is effect of FeN₄O₂-SACs in psoriasis relapse. Data from TRM cells are not convincing. Authors show 70-80% of CD69+ CD103+ from the CD3+ population under control condition which is not in line with images of Figure 6g. Murine resident memory T cells are mainly at epidermis. Identification of TRM cells should be done using immunofluorescence using anti-CD8 and anti-CD103. J Invest Dermatol. 2020 Apr; 140(4): 748–755.

Response: Thank you for your valuable suggestions. Methodologically, flow cytometry has higher

sensitivity and accuracy than immunohistochemistry (IHC). Flow cytometry excels in the quantitative analysis of individual cells in suspension, while IHC provides spatial information about protein expression within tissue sections. The qualitative information about the presence of CD103 in tissue sections is provided by IHC (Figure 6g,h). The flow cytometry results exhibit higher levels of CD103⁺ cells than those in the IHC group in the normal cohort, which is likely due to the fact that our flow cytometry assessment of CD103⁺ cells was conducted within the CD3 subpopulation, resulting in a relatively enhanced proportion of CD103⁺ cells.

Upon considering your suggestion, we re-analyzed our data, which shows the presence of differences in CD8⁺CD103⁺ tissue-resident memory (TRM) cell proportions among different groups, which is consistent with the observed data in the IHC results. Additionally, the reference you mentioned is very valuable and has been cited as ref [59].

The following is our revisions to the manuscript:

On Page 5 of Revised Supplementary Information, which reads:

Flow cytometry

The isolation of cells from the skin tissues of mice was made following the previous method². In flow cytometry, cells were stained with fluorescently labeled antibodies directly after cell processing. For experiments with murine tissues, the following mAbs were used for extracellular staining: CD45-BV421(30-F11, Biolegend, 103134), CD3-FITC (17A2, Biolegend, 100204), CD4-PerCP (RM405, Biolegend, 100538), CD8-PE (53-6.7, Biolegend, 100708), F4/80-APC (BM8, Invitrogen, 17-4801-82), CD11b-BV785 (M1/70, Biolegend, 101243), Viability-APC-Cy7 (Invitrogen, 65-0865-14), and CD103-BV510 (2E7, Biolegend, 121423). Samples were acquired on the BD LSR Fortessa X-20 equipped with four lasers (BD Biosciences) and data were analyzed using FlowJo software (Ashland, OR).

On page 15 of Revised Manuscript, which reads:

Consistently, the flow cytometry results shows that FeN₄O₂-SACs remarkably reduce the CD3⁺T cells, F4/80⁺CD11b⁺macrophages and CD103⁺CD8⁺T_{RM} cells (Supplementary Fig. 26-27)⁵⁹.

And on page 32 of Revised Supplementary Information as Supplementary Fig. 26, which is:

Supplementary Fig. 26. FeN₄O₂-SACs remarkably reduce the numbers of CD3⁺ T lymphocytes, CD11b⁺F4/80⁺ macrophages and CD103⁺CD8⁺T_{RM} cells at the ear skins. a, The percentage of CD3⁺ T lymphocytes in different groups determined by flow cytometry. n=6 samples/group. **b**, The percentage of CD11b⁺F4/80⁺ macrophages (in CD45⁺ cells) in different groups quantified by flow cytometry. n=6 samples/group. **c**, The percentage of CD103⁺CD8⁺T_{RM} cells (in CD3⁺ T lymphocytes) in different groups quantified by flow cytometry. n=5-6 samples/group. ****p*<0.001, ***p*<0.01 versus the IMQ group; #*p*<0.05 versus the IMQ+Cal group; ■■■*p*<0.001 versus the NC group.

3. How long does the antioxidant effect of the compound last once it has been applied to the skin?

Response: We sincerely appreciate your thoughtful comment. The ROS levels at the mouse skin of both the IMQ+FeN₄O₂-SACs group and the Relapse+FeN₄O₂-SACs group were remarkably decreased (Fig. 6j,k and Supplementary Fig. 36) compared to the negative control. The antioxidant effect maintained for 40 days after the FeN₄O₂-SACs were discontinued in the latter group. Thus, once FeN₄O₂-SACs were applied to the skin, the effective antioxidation period of the compound is at least

40 days.

4. Supplementary figure 21. Panel a, Y axis indicate proliferative level but cell viability, authors quote at figure legend and results that this figure correspond to cytotoxicity level, please clarify this point.

Response: We truly appreciate your insightful comments. In this study, the cell viability/proliferative level was detected using CCK-8 assay, which presents the activity of cells, as well as cell proliferation. In Supplementary Figure 21, Panel a should indicate the cell viability rather than proliferative level, thus the Y axis has been changed into “cell viability” to avoid possible confusions as you suggested.

The following is our revisions to the manuscript:

On Page 27 of Revised Supplementary Information as Supplementary Fig. 21, which is:

Supplementary Fig. 21. FeN₄O₂-SACs effectively inhibit the hyperproliferation and inflammation of NHEK cells. a, The cytotoxicity profile of FeN₄O₂-SACs on NHEK cells (48 h). Data from 3 separate experiments as mean ± SD.

5. Supplementary figure 24. Gating strategy should be included to demonstrate that death cell are excluded from the analysis. Also, M5 instead IL-17 should indicated.

Response: Thank you for the professional comments. As suggested, the ROS levels induced by M5 were measured in normal human epidermal keratinocytes (NHEK cells), and the results including the gating strategy are depicted below (Supplementary Fig 24). Death cells were excluded from the analysis by staining with the cell viability dye propidium iodide (PI).

The following is our revisions to the manuscript:

On Page 30 of Revised Supplementary Information as Supplementary Fig. 24, which is:

Supplementary Fig. 24. Analysis of ROS levels in NHEK cells. **a,b**, Flow cytometry analysis of NHEKs cells in different groups stained by DCFH-DA (2,7-Dichlorodihydrofluorescein diacetate, EX498/EM530 nm) (**a**) and the quantification (**b**) of fluorescence intensity. Data from 3 separate experiments as mean ± SD. *** $p < 0.001$ versus the Control group; ### $p < 0.001$ versus the M5 group. **c**, Gating strategy to determine the intracellular ROS level presented in supplementary Fig. 24b.

6. Figure 5. Please include gating strategy, it is important to confirm that cell viability dye has been used to exclude death cells from the analysis

Response: Thank you very much for pointing out this issue. The cell viability dye PI was employed to exclude death cells, and the results including the gating strategy have been supplemented in Figure 5 and Supplementary Figure 22.

The following is our revisions to the manuscript:

On Page 14 of Revised Manuscript, which is:

Fig. 5. b,c, Flow cytometry analysis of HaCaT cells in different groups stained by DCFH-DA (2,7-Dichlorodihydrofluorescein diacetate, EX498/EM530 nm) (b) and the quantification of fluorescence intensity (c). Data from 3 separate experiments as mean \pm SD.

On page 26 of Revised Manuscript, which reads:

***In vitro* ROS detection**

ROS assay Kit (50101ES01, Yeason) was utilized to detect the ROS level in HaCaT cells according to the manufacturer’s instruction. In Brief, cells in different groups were treated with blank (Control), FeN₄O₂-SACs (2.5 μ g/ml), M5, and M5+FeN₄O₂-SACs for 48 h, respectively. DCFH-DA was diluted with serum-free medium and DCFH-DA was added to the cell suspension and incubated for 30 min at 37°C in the dark. The fluorescence images were visualized and recorded using a fluorescence microscope at 498/530 nm (excitation/emission) and \times 200 magnification.

In flow cytometry analysis, propidium iodide solution (Biolegend, 421301) was added and incubated for 5 min to dye the dead cells. The ROS levels in normal human epidermal keratinocytes (NHEK cells) were also measured using the same procedure.

And on page 28 of Revised Supplementary Information as Supplementary Fig. 22, which is:

Supplementary Fig. 22. Cell gating strategy to determine the intracellular ROS level presented in Fig.

5c.

7. Supplementary figure 27. It is mandatory to include a cell viability dye for these assays.

Response: Thank you for your thoughtful comments. The cell viability dye for these assays were included and shown as below. Cells were stained with viability dye (APC-cy7) and cell viability was evaluated by flow cytometry. The results indicate no significant differences in the cell viability among different groups.

Supplementary Fig. 27: The comparison of cells viability among different groups quantified by flow cytometry. n=6 samples/group.

We hope these revisions address your concerns. Thanks again for your kind help to improve the manuscript. All these efforts may, as we hope, make the article suitable for publication in *Nature Communications*.

Reviewers' Comments:

Reviewer #3:

Remarks to the Author:

I'm aware the authors efforts to address my concerns. However, I remain concerned regards the two points. The first one is the death of up to 50% of the cells after 48h of incubation, which could lead to misinterpretation of the in vitro results. I requested to include the gating strategy from Figure 5, not just from panel 5c. Proliferation analysis should be done on live cells and it is necessary to demonstrate that this was the case using a fixable viability dye. Also, in the new supplementary figures 22 and 24 the number of dead cells do not correspond with the 50% showed in supplementary figure 19. The second one is the effect of FeN4O2-SACs on TREM cells. In the new supplementary figure 26, plot for cell viability is missing. It is very important to include it because the diagonal image of CD8+ CD103+ cells strongly suggest dead cells.

As a minor comment, please revise carefully figure and figure legends, there is an error in figure 8b, symbols (* and #) are wrong indicated, I guess that in both cases psoriasis group and relapsed group the comparison is vs control group. In addition, figure legend state "vs IMQ group" (page 23 line 443). In figure legend 5 the incubation time for data in panel "e" is missing. I am sorry I did not see this small error in the latest version.

Response to reviewer #3.

Reviewer #3 (Remarks to the Author):

I'm aware the authors efforts to address my concerns. However, I remain concerned regards the two points.

Response: Thank you very much for the professional suggestions. Please find the following detailed responses to your suggestions.

1.The first one is the death of up to 50% of the cells after 48h of incubation, which could lead to misinterpretation of the in vitro results. I requested to include the gating strategy from Figure 5, not just from panel 5c. Proliferation analysis should be done on live cells and it is necessary to demonstrate that this was the case using a fixable viability dye. Also, in the new supplementary figures 22 and 24 the number of dead cells do not correspond with the 50% showed in supplementary figure 19.

Response: Thank you for your thorough review and professional comments. Firstly, we carefully checked the source data and we apologize for any confusion caused by Supplementary Fig.19. In the previous version, it may not have been clearly conveyed that this figure presents the concentrations on a logarithm scale (log10). To address this, we have now redrawn Supplementary Fig. 19 to ensure its legibility. As shown in the revised figure, the IC₅₀ value of FeN₄O₂-SACs is 87.8 ug/ml, rather than 2.5 ug/ml, to result in 50% cell death when incubated for 48 h.

Secondly, regarding Figure 5, the gating strategies apply only to Fig. 5b and Fig. 5c, as shown in Supplementary Fig. 21. While Fig. 5d and Fig. 5e, on the other hand, present CCK-8 and inflammatory factor assays, respectively. In the proliferation analysis, we used the CCK-8 (Fig. 5d) and EdU assays (Supplementary Fig. 20). In the EdU analysis, EdU was embedded in newly synthesized DNA during cell proliferation and was subsequently labeled by the biotin probe; whereas dead cells lack the ability to synthesize new DNA and thus cannot be labeled. Therefore, it was the proliferating viable cells that were detected by the EdU assay.

Thirdly, the gating strategy shown in Supplementary Fig. 22 and 24 represents the Control group, hence no significant cell death is shown in the figures. Similarly, there is no significant cell death in

the other groups. We have modified the figure legends to clearly demonstrate the group depicted by the gating strategy. We appreciate your professional input and thank you once again for your professional comments.

The following is our revisions to the manuscript:

Revised Supplementary Fig. 19 on Page 25 in Supplementary Information:

Supplementary Fig. 19. Cytotoxicity of FeN₄O₂-SACs against HaCaT cells in 48 h of incubation.

Revised Supplementary Fig. 22 on Page 28 in Supplementary Information:

Supplementary Fig. 22. Gating strategy (presented using the Control group) to determine the intracellular ROS level presented in Fig. 5c.

Revised Supplementary Fig. 24 on Page 30 in Supplementary Information:

Supplementary Fig. 24. Analysis of ROS levels in NHEK cells. c. Gating strategy (presented using the Control group) to determine the intracellular ROS level presented in supplementary Fig. 24b.

2. The second one is the effect of FeN4O2-SACs on TRM cells. In the new supplementary figure 26, plot for cell viability is missing. It is very important to include it because the diagonal image of CD8+CD103+ cells strongly suggest dead cells.

Response: Thank you for the constructive suggestion. In response to your query, we have supplemented the gating strategy applied to CD8+CD103+ TRM cells (Supplementary Fig. 27a). Moreover, we have included flow cytometry plot images in Supplementary Fig. 27b to clarify the ratio of CD8+CD103+ TRM within the viable cell population, thus providing additional support for their viability. We appreciate your meticulousness and attention to the important details.

The following is our revisions to the manuscript:

Revised Supplementary Fig. 27 on Page 33 in Supplementary Information:

Supplementary Fig. 27. a, Gating strategy applied to CD8⁺CD103⁺ TRM cells in Supplementary Fig. 26c. **b,** Flow cytometry plots of live CD8⁺CD103⁺ TRM cells among live cells.

3. As a minor comment, please revise carefully figure and figure legends, there is an error in figure 8b, symbols (* and #) are wrong indicated, I guess that in both cases psoriasis group and relapsed group the comparison is vs control group. In addition, figure legend state "vs IMQ group" (page 23 line 443). In figure legend 5 the incubation time for data in panel "e" is missing. I am sorry I did not see this small error in the latest version.

Response: Thank you for your meticulous review. The symbols (* and #) in Fig. 8b and the corresponding Figure legend have been corrected. It is now clear that both the psoriasis group and the relapsed group were compared with the Control group. Furthermore, we have supplemented the Figure legend to indicate that the incubation period of 24 hours for the data presented in Fig. 5e. In addition, we have carefully checked the rest of the Figures and Figure legends, ensuring their accuracy. We appreciate your reminder and thank you once again for your valuable comments.

The following is our revisions to the manuscript:

Revised Fig. 8 on Page 23 in revised manuscript:

Fig. 8. b, Relative quantifications of CD3 and ESR1 determined by ImageJ. n=4 samples/group. *** $p < 0.001$ versus the Normal group.

And Revised Fig. 5 on Page 14 in the revised manuscript:

Fig 5. e, RT-qPCR results for the inflammatory factors *TNF- α* , *IL-6*, and *IL-8* at 24 hours.

We hope these revisions can fully address your concerns. Thanks again for your kind help to improve the manuscript. All these efforts may, as we hope, make the article acceptable for publication in *Nature Communications*.

Reviewers' Comments:

Reviewer #3:

Remarks to the Author:

Authors have addressed all my concerns, I don't have additional comments, just the recommendation to replace figures of gating strategy using data of treated samples instead of control group. It is important for readers to have these data.

Response to reviewer #3.

Reviewer #3 (Remarks to the Author):

Authors have addressed all my concerns, I don't have additional comments, just the recommendation to replace figures of gating strategy using data of treated samples instead of control group. It is important for readers to have these data.

Response: Thank you very much for the positive comment and professional recommendation. The gating strategy has been revised using data of treated samples.

The following is our revisions to the manuscript:

Revised Supplementary Fig. 22 on Page 28 in Supplementary Information:

Supplementary Fig. 22. Gating strategy (presented using the M5 group) to determine the intracellular ROS level presented in Fig. 5c.

Revised Supplementary Fig. 24 on Page 30 in Supplementary Information:

Supplementary Fig. 24. Analysis of ROS levels in NHEK cells. c, Gating strategy (presented using the M5 group) to determine the intracellular ROS level presented in supplementary Fig. 24b.

Thanks again for your time and effort, which are highly helpful in improving the quality of the manuscript. All these efforts may, as we hope, make the article acceptable for publication in *Nature Communications*.